# Large and increasing methane emissions from Eastern Amazonia derived from satellite data, 2010 - 2018

Chris Wilson[1,2], Martyn P. Chipperfield[1,2], Manuel Gloor[3], Robert J. Parker[4,5], Hartmut Boesch[4,5], Joey McNorton[6], Luciana V. Gatti[7], John B. Miller[8], Luana S. Basso[7], Sarah A. Monks[9,10,*]

[1]National Centre for Earth Observation, University of Leeds, Leeds, UK
[2]School of Earth and Environment, University of Leeds, UK
[3]School of Geography, University of Leeds, Leeds, UK
[4]Earth Observation Science, School of Physics and Astronomy, University of Leicester, Leicester, UK.
[5]National Centre for Earth Observation, University of Leicester, Leicester, UK
[6]European Centre for Medium-Range Weather Forecasts, Reading, UK
[7]Earth System Science Center (CCST), National Institute for Space Research (INPE), Av. Dos Astronautas, 1758, 12.227-010, São José dos Campos, SP, Brazil
[8]Global Monitoring Laboratory, National Oceanic and Atmospheric Administration, Boulder, Colorado, USA
[9,*]Formerly at CIRES, University of Colorado Boulder, Boulder, CO, USA
[10,*]Formerly at Chemical Sciences Division, NOAA, Earth System Research Laboratory, Boulder, CO, USA

*Correspondence to*: Chris Wilson (c.wilson@leeds.ac.uk)

**Abstract.** We use a global inverse model, satellite data and flask measurements to estimate methane ($CH_4$) emissions from South America, Brazil and the basin of the Amazon river for the period 2010 – 2018. We find that emissions from Brazil have risen during this period, most quickly in the Eastern Amazon basin, and that this is concurrent with increasing surface temperatures in this region. Brazilian $CH_4$ emissions rose from $49.8 \pm 5.4$ Tg yr$^{-1}$ in 2010 – 2013 to $55.6 \pm 5.2$ Tg yr$^{-1}$ in 2014 – 2017, with the wet season of December – March having the largest positive trend in emissions. Amazon basin emissions grew from $41.7 \pm 5.3$ Tg yr$^{-1}$ to $49.3 \pm 5.1$ Tg yr$^{-1}$ during the same period. We derive no significant trend in regional emissions from fossil fuels during this period. We find that our posterior distribution of emissions within South America is significantly and consistently changed from our prior estimates, with the strongest emission sources being in the far north of the continent and to the south and south-east of the Amazon basin, at the mouth of the Amazon river and nearby marsh, swamp and mangrove regions. We derive particularly large emissions during the wet season of 2013/14, when flooding was prevalent over larger regions than normal within the Amazon basin. We compare our posterior $CH_4$ mole fractions, derived from posterior fluxes, to independent observations of $CH_4$ mole fraction taken at five lower- to mid-tropospheric vertical profiling sites over the Amazon and find that our posterior fluxes outperform prior fluxes at all locations. In particular the large emissions from the eastern Amazon basin are shown to be in good agreement with independent observations made at Santarém, a location which has long displayed higher mole fractions of atmospheric $CH_4$ in contrast with other basin locations. We show that a bottom-up wetland flux model can neither match the variation in annual fluxes, nor the positive trend in emissions, produced by the

inversion. Our results show that the Amazon alone was responsible for $24 \pm 18\%$ of the total global increase in $CH_4$ flux during the study period, and it may contribute further in future due to its sensitivity to temperature changes.

## 1 Introduction

Methane ($CH_4$), a strong greenhouse gas emitted from a variety of anthropogenic and natural sources, is second only to carbon dioxide ($CO_2$) in its importance regarding the anthropogenic radiative forcing contributing to Earth's climate change (Myhre et al., 2013). Approximately 90% of the $CH_4$ that is emitted into the atmosphere is eventually destroyed through reaction with the hydroxyl (OH) radical and most of the remainder is lost to other, smaller sinks, but a net positive imbalance means that the atmospheric burden of $CH_4$ has been increasing steadily since preindustrial times (e.g. Rubino et al., 2019). With an atmospheric lifetime of approximately 9 years (Prather et al., 2012), $CH_4$ is a potentially important species for short-term gains in mitigation of anthropogenic climate change (Shindell et al., 2012). However, the magnitude of the total global sources and sinks of $CH_4$ are still not well quantified (Saunois et al., 2020). The geographical distribution and sectoral attribution of methane emissions, and the inter-annual variation of these sources, are also uncertain (Saunois et al., 2016; Schaefer, 2019). This leads to difficulties in assessing potential emission mitigation strategies, hampering our ability to meet and assess the criteria for limiting the global temperature increase put forward as part of the Paris climate agreement (Nisbet et al., 2019).

The atmospheric methane burden is now approximately 2.5 times larger than it was in 1750 (Rubino et al., 2019). The global mean burden stabilised between 2000 and 2006, after which it began increasing again (Nisbet et al., 2016). Concerningly, the rate of increase of the atmospheric burden has accelerated since 2014 (Nisbet et al., 2019). This suggests that $CH_4$ emissions have been increasing at an accelerated rate during the past decade, but our understanding of how emissions are changing is complicated by the following:

(1) attributing a potential emission increase to a particular region and/or sector is complex, leading to conflicting hypotheses regarding the changing fluxes (e.g. Nisbet et al., 2016; Worden et al., 2017; Monks et al., 2018; Schaefer, 2019; Lan et al., 2019; Jackson et al., 2020). Indeed, whilst rising atmospheric mole fractions of greenhouse gases usually signifies increasing anthropogenic influence, the changing isotopic signature of atmospheric $CH_4$ as the burden rises initially appears to indicate that fossil fuel emissions might not be the main driver for the increase (Schaefer et al., 2016; Nisbet et al., 2019; Fujita et al., 2020). Other sectors, including anthropogenic agricultural emissions, could be responsible. On the other hand, it has been argued that increasing fossil fuel emissions could still be reconciled with the observed isotopic signature, along with increasing biogenic fluxes, if emissions from fires have decreased during the same period (Worden et al., 2017; Thompson et al., 2018; Howarth, 2019; Chandra et al., 2021); and,

(2) the uncertainty surrounding the distribution and variation of tropospheric OH means that variations, or negative trends, in this major atmospheric sink of methane might also have played some role in the stabilisation and renewed rise (McNorton et

al., 2016; Rigby et al., 2017; Turner et al., 2017; McNorton et al., 2018). However, others have found no significant trend in OH during this period (e.g. Naus et al., 2020; Patra et al., 2021), or even a trend in the opposing direction (Zhao et al., 2020).

In general, anthropogenic emissions of $CH_4$ from fossil fuels, agriculture and waste are better constrained than natural emissions, particularly in bottom-up inventories (Saunois et al., 2020). The majority of natural emissions come from wetlands, with smaller contributions from inland freshwaters, oceans, termites, wild animals and geological seeps. There are also small but significant emissions from biomass burning, which are sometimes counted separately from other anthropogenic emissions despite often being due to agricultural land clearing (van der Werf et al., 2017).

Wetlands are the largest single-sector contributors to the global methane flux (Saunois et al., 2020) and the basin of the Amazon river in South America, covering an area of approximately 6,000,000 $km^2$ (Poulter et al., 2010), is a significant contributor to the global wetland $CH_4$ emission budget (Wilson et al., 2016; Bloom et al., 2017). Approximately 60% of the basin is within the borders of Brazil. Wetland regions within Amazonia generally include seasonal floodplains in the east and swamps, bogs and marsh regions in the west, along with areas of mangroves along parts of the coast. Throughout the rest of this study, we group all of these distinct ecosystems together as 'wetlands' for brevity. As well as a number of large wetland sources within South America, there are often significant contributions from fires during warmer, drier years (van der Werf et al., 2017). Recent studies have suggested that there is also a direct contribution of fluxes entering the atmosphere via trees in the Amazon, although there are likely some cases of this flux having been already included as part of the wetland flux in some inventories (Pangala et al., 2017). In fact, the contribution of each of these sources of $CH_4$, along with their regional distribution and variability over time, is still relatively uncertain. In studies published in the 2000s and early 2010s, estimates of $CH_4$ emissions from the Amazon basin ranged from 4 to 92 $Tg(CH_4)$ $yr^{-1}$ (henceforth Tg $yr^{-1}$, Melack et al., 2004; do Carmo et al., 2006; Miller et al., 2007; Kirschke et al., 2013), but recently estimates have converged somewhat, e.g. 31.6 – 41.1 Tg $yr^{-1}$ (Wilson et al., 2016), 42.7 ± 5.6 Tg $yr^{-1}$ (including tree flux, Pangala et al., 2017) and 44.4 ± 4.8 Tg $yr^{-1}$ (Ringeval et al., 2014). The global wetland total was recently estimated to be 148 ± 25 Tg $yr^{-1}$ from bottom-up estimates and 159 – 200 Tg $yr^{-1}$ from top-down models (Saunois et al., 2020), which implies that if the majority of the emissions from the Amazon are from wetlands, then the region contributes up to ~30% of the global $CH_4$ wetland flux.

Many studies have attempted to estimate national $CH_4$ emissions rather than from ecosystems such as the Amazon, partly as it will likely be easier for countries to put in place emission reduction protocols on a national basis. Some recent studies have therefore reported emission totals for the country of Brazil. The synthesis of Saunois et al. (2020) used a suite of top-down models to find a wide range of 47.3 – 78.2 Tg $yr^{-1}$ total emissions from all sources within Brazil during the period 2008 – 2017. Natural sources made up 26.9 – 53.8 Tg $yr^{-1}$ of this total. Janardanan et al. (2020) used a global inversion to constrain total Brazilian emissions to 56.2 ± 10 Tg $yr^{-1}$ in the period 2011-2017. However, Tunnicliffe et al. (2020) used a high-resolution regional inversion to find much smaller emissions from the country, calculating total Brazilian emissions of 33.6 ± 3.6 Tg yr⁻

[1], with wetlands making up $13.0 \pm 1.9$ Tg yr$^{-1}$ of this total. The relatively large range of estimates produced by these studies, some of which make use of the same observational datasets, is indicative of the difficulties inherent in using top-down methods to assess surface emissions of CH$_4$ from within the poorly monitored continent of South America. However, in order to best understand the global methane budget and its sources, it is still vital that the significant contribution of South American emissions is evaluated and attributed.

In order to best unite these estimates, regular observation of atmospheric methane over South America is necessary. The Thermal And Near infrared Sensor for carbon Observations – Fourier Transform Spectrometer (TANSO-FTS) instrument on the GOSAT satellite (Kuze et al., 2009) is particularly advantageous, as it is sensitive far down into the troposphere and has been providing regular global coverage of atmospheric CH$_4$ continuously since April 2009 (Parker et al., 2020a). This decade of uninterrupted global coverage allows for understanding of methane variations over a much longer period than many of the other available datasets, particularly in the tropics.

In this paper we use CH$_4$ observations from GOSAT along with flask measurements both from within and outside the Amazon basin to provide an almost complete 10-year record of methane emissions from South America, beginning in 2009. We use the TOMCAT chemical transport model and its inverse model, INVICAT, to quantify emissions and their uncertainties during this decade. Ours aims are to 1) assess the geographical distribution of South American CH$_4$ emissions, with focus on the country of Brazil and the Amazon basin ecosystem; 2) examine how these emissions have changed during the previous decade; and 3) investigate why any changes to natural emissions might have occurred. We describe the observations used and the modelling methodology in Section 2. We show our results and discuss our findings in Section 3 and Section 4, respectively.

## 2 Methods

### 2.1 Observations

We assimilate both in-situ flask observations and GOSAT satellite retrievals of CH$_4$ into the inverse model. We also use, but do not assimilate, a set of observations made as part of regular flask-based aircraft monitoring campaign within the Amazon basin since 2010 for validation of our results.

### 2.1.1 Surface flask observations

We assimilate global long-term surface data of CH$_4$ provided by the National Oceanic and Atmospheric Administration's Global Monitoring Laboratory (NOAA GML, Table A1). We use data from 56 background monitoring sites, the locations of which are shown in Figure 1. Whole air samples in flasks are collected weekly to biweekly at each site, and CH$_4$ is measured using gas chromatography with a flame ionization detection method (Dlugokencky et al., 2018). Data from these sites are

assimilated in order to constrain the background variations in $CH_4$ mole fractions at the Earth's surface. The observations made

at these locations have high accuracy but are generally located in regions that are not near significant sources of $CH_4$. There is

a relative lack of regular observation in tropical regions, where $CH_4$ emissions are significant and uncertain. These observations

can therefore provide accurate values for background $CH_4$ mixing ratios but are not usually able to provide accurate regional

$CH_4$ distributions in those areas that require the most constraint.

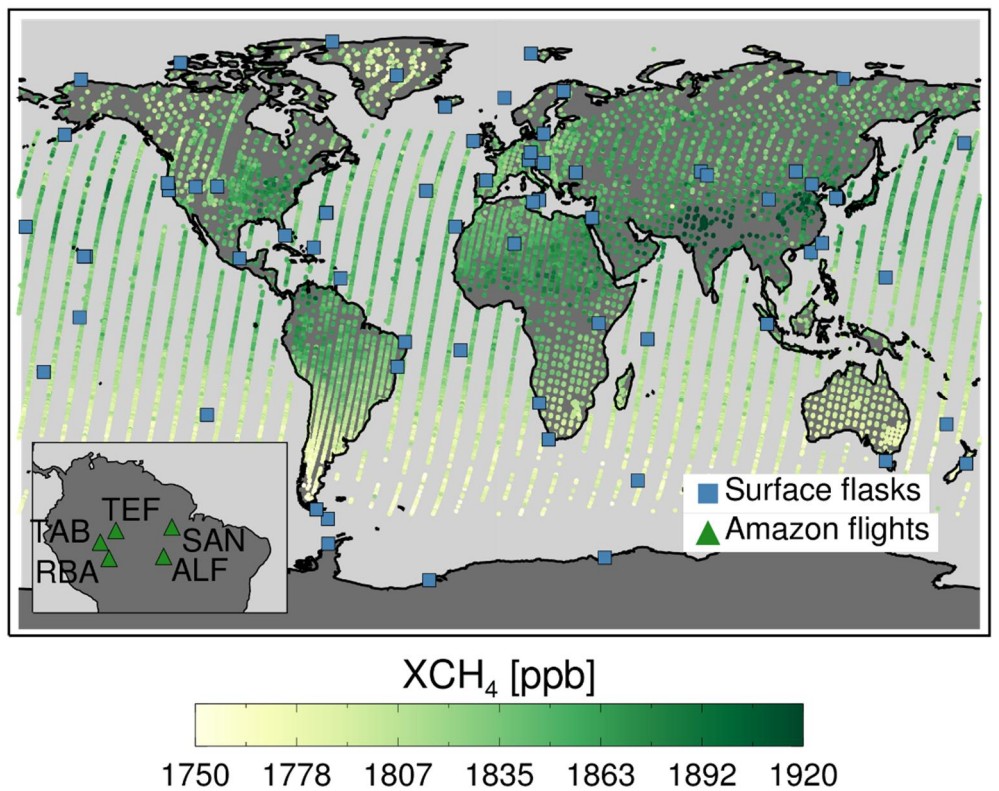

**Figure 1: Locations of NOAA surface sites from which flask-based measurements of $CH_4$ are assimilated (blue squares), along with locations and values of GOSAT $XCH_4$ retrievals for August 2017 (circles). Inset shows locations of flight-based observations of $CH_4$ within the Amazon basin (green triangles).**

**2.1.2 GOSAT observations**

We also assimilate column-averaged dry-air mole fractions of $CH_4$ ($XCH_4$) from the University of Leicester Proxy retrieval

scheme v7.2 for GOSAT (Parker et al., 2011, 2020a). This dataset has been used in the past in forward modelling studies to

assess wetland $CH_4$ emissions using the TOMCAT model (Parker et al., 2018, 2020b). The GOSAT Proxy scheme uses the

ratio of the retrieved $XCO_2$ and $XCH_4$, together with model-based estimates of $XCO_2$, in order to reduce the effects of

atmospheric scattering and improve coverage of $XCH_4$ retrievals. This is particularly true in tropical land regions where the

prevalence of cloudy pixels often restricts the successful direct retrieval of XCH4. GOSAT XCH4 retrievals have been used previously in a number of forward and inverse modelling studies (Fraser et al., 2013; McNorton et al., 2016; Feng et al., 2017; Miller et al., 2019). The observations are regularly validated against independent data, including CH4 observations made as part of the Total Carbon Column Observing Network (TCCON, Wunch et al., 2011), although none of the measurement sites

included as part of this network are located within the Amazon region. Webb et al. (2016) compared GOSAT XCH4 to vertical profiles of flask-based measurements of CH4 taken at a number of sites within the Amazon basin (described here in Section 2.1.3) and found that biases between the satellite retrievals and the flask observations were not significantly different from zero.

Before assimilation, GOSAT retrieved and *a priori* XCH4 were averaged onto the model grid. Both sun-glint observations over the oceans and nadir observations over land were included in the inversion. All XCH4 values measured by the satellite during one model timestep in the same grid cell were averaged using a weighted mean according to their uncertainties. The largest number of observations combined into a single value was 32, and the mean number was 4.7 over land and 6.0 over oceans. Within the Amazon basin, the mean number of observations combined was 3.8. Figure 1 shows an example monthly

distribution of observations used in the inversion. For accurate comparison between the retrieved XCH4 and those simulated by the model, GOSAT averaging kernels falling in the same model grid cell and time step were averaged, similarly to the XCH4, and applied to the model vertical profile. Using a single model profile in each grid cell and model time step allows the use of averaging kernels that have been averaged in this way without introducing a bias, due to the distributive property of matrix multiplication. Retrievals where the model and satellite surface pressure differed by more than 50 hPa were rejected.


Due to a range of potential error sources in both the atmospheric transport model and the GOSAT retrievals, there is a persistent bias between them, which varies with latitude. We quantified this bias by comparing the results of a previous inversion, in which only the surface flask observations had been assimilated for the full 2009-2018 period, to the GOSAT XCH4. We applied the averaging kernels to the three-dimensional (3-D) CH4 output from the flask data inversion and calculated the model –

observation zonal mean bias $B(\varphi)$, in parts per billion (ppb), as a function of latitude ($\varphi$), over this period:

$$B(\varphi) = 0.0016\varphi^2 - 0.1\varphi + 4.4 \,, \tag{1}$$

where $\varphi$ is equal to the latitude of the observation in degrees north (see Figure A1). Positive values of $B(\varphi)$ indicate positive observation bias relative to the model. Including a function that is constant along the longitudinal and temporal axes means that all information content from the satellite data along these axes is preserved, but this method reduces conflict between

assimilation of the satellite and flask observations. Similar methods have been used before, for example in Bergamaschi et al. (2009). Across the tropics (30°S – 30°N), the derived bias varies between 2.8 and 8.8 ppb. Further south, the bias reaches values up to 13.4 ppb. In our analysis we add the estimated bias value to the simulated XCH4 values in the inversion after the averaging kernels are applied.

### 2.1.3 Amazonian aircraft profiles

We used independent in situ observations of $CH_4$ mole fraction made within the basin to validate our inversion results. Since 2010, aircraft-borne flask air observations of a number of species, including $CH_4$, $CO_2$ and carbon monoxide (CO) have been made at five locations within the Amazon basin (shown in Figure 1, inset) by researchers at the Instituto de Pesquisas Energéticas e Nucleares (IPEN) in Sao Paulo, Brazil until 2014 and at the National Institute for Space Research (INPE) Sao Jose dos Campos, Brazil (2015 – present). The sites are located at Santarem (SAN, 55.0°W, 2.9°S), Tabatinga (TAB, 69.7°W, 6.0°S), Alta Floresta (ALF, 56.7°W, 8.9°S), Rio Branco (RBA, 67.9°W, 9.3°S) and Tefé (TEF, 66.5°W 3.6°S). Measurements were only ever made concurrently at four locations, as the measurements at Tefé were started in 2013, to replace those made at Tabatinga up to 2012. We therefore combine observations made at these locations and refer to them as TAB/TEF throughout this manuscript. Both sites are located in the north-west of the Amazon basin and sample similar incoming air masses. Flights are undertaken at approximately biweekly intervals above each site up to an altitude of ~4.4 km, and 0.7 L flasks were filled every 300–500 m to produce vertical profiles. All measurements were taken between 12:00 and 13:00 local time, when the boundary layer is fully developed. The flasks were analysed for $CH_4$ mole fractions at the high-precision gas analytics laboratory at IPEN and INPE, following the NOAA GML approach, including rigorous calibration to the World Meteorological Organization (WMO) $CH_4$ mole fraction scale. The measurement locations were chosen in order to sample the dominant tropospheric airstream across the basin. These observations were not assimilated in the inversion. For more information about these measurements, see Gatti et al. (2014) and Basso et al. (2016).

## 2.2 Model set-up

### 2.2.1 Inverse model set-up

The TOMCAT model is a global 3-D Eulerian offline chemical transport model (CTM) (Chipperfield, 2006; Monks et al., 2017). It has been used in a number of previous studies of atmospheric composition and transport (e.g. Wilson et al., 2016; McNorton et al., 2016; Parker et al., 2018). We use the INVICAT inversion framework (Wilson et al., 2014), which is based on the TOMCAT model and its adjoint. INVICAT uses a variational scheme based on 4D-Var methods used in Numerical Weather Prediction (NWP) and has been used in the past to constrain emissions of species including $CO_2$, ethane ($C_2H_6$), and nitrous oxide ($N_2O$) (Gloor et al., 2018; Monks et al., 2018; Thompson et al., 2019; Tian et al., 2020). The inverse method employed by INVICAT is described in depth in these previous references. In brief, the method aims to minimise, in a least-squares sense, the value of a cost function. The cost function is an error-weighted sum of the model – observation mismatch, plus error-weighted departures from the *a priori* flux estimate.

The inversion input is in the form of an *a priori* mean flux value for each grid cell along with an error covariance matrix for these values, and the output is an *a posteriori* mean grid cell flux value and error covariance matrix. Mean *a priori* and *a*

*posteriori* atmospheric mole fractions of CH$_4$ are also produced. For brevity, throughout the remainder of this text, we will refer to the mean *a priori* and *a posteriori* fluxes as 'prior fluxes' and 'posterior fluxes', respectively. Similarly, the mean *a priori* and *a posteriori* mole fractions will be referred to as prior and posterior mole fractions.

The forward and adjoint model simulations were carried out at 5.6° × 5.6° horizontal resolution, with 60 vertical levels up to 0.1 hPa. The model time step was 30 minutes. The meteorology was taken from the European Centre for Medium-Range Weather Forecasts (ECMWF) ERA-Interim reanalyses (ERA-I, Dee et al., 2011). The inversions were carried out for each year separately and each completed 40 minimisation iterations. For each year's inversion, 40 iterations were enough for the cost function and its gradient norm to be judged to have converged (less than 1% variation through 5 consecutive
iterations). The inversion for each year was actually run for 14 months up to the end of February for the following year, with the final two months being discarded from the results. This was in order to better constrain fluxes during the final months of each year. Each inversion therefore overlapped with the following one for two months but was initialized using 3-D fields provided from the correct date in the previous year, so that total CH$_4$ burden was conserved across years.

For the assimilated surface observations, the model output was linearly interpolated to the correct longitude, latitude and altitude, at the nearest model timestep. For the averaged GOSAT observations, the model mole fractions were interpolated to the correct longitude and latitude at the nearest time step, before the GOSAT averaging kernels were applied to the model output to give an XCH$_4$ value comparable with GOSAT. GOSAT observations are given an uncorrelated uncertainty equal to 2.5 times the supplied retrieval error, which ranged from 3.5 to 25.8 ppb, in order to account for representation error and
observation correlations removed by the averaging of the retrievals, as in Chevallier (2007). This inflation value was based on the mean number of observations combined in each grid cell. In short sensitivity tests, the magnitude of posterior emissions was not sensitive to this inflation factor once it was larger than 2, although the posterior error estimate was affected. This choice gave a mean GOSAT XCH$_4$ uncertainty value of 24.4 ppb. NOAA observations are given uncorrelated errors of 3 ppb plus representation error. For these observations, representation error was estimated as the mean difference across the 8 grid
cells surrounding the cell containing the observation location.

Prior emissions are given grid cell uncertainties of 250% of the prior flux value, but also included spatial and temporal correlations. Although inversions such as this do not directly allow for sectorial analysis of emissions, we use the off-diagonal values of the prior covariance matrix to provide some information of this nature. Similar to Meirink et al., (2008), we split our
prior and posterior solutions into the anthropogenic fossil fuel emissions assumed to be strongly correlated in time (FF), and emissions with strong seasonal cycles from natural, agricultural and biomass burning sources (NAT + AGR + BB) by imposing prior temporal correlations on the FF contributions. FF emissions in each grid cell in each month were correlated with emissions from the same grid cell in other months with an exponential correlation function with a time scale of 9.5 months (equivalent to a consecutive-month correlation of 0.9). Both NAT + AGR + BB and FF sectors had spatial correlations imposed

between grid cells, based on Gaussian covariance functions with correlation length scales of 500km. This gives global uncertainty of approximately 70 Tg yr$^{-1}$. The sectors which make up the NAT + AGR + BB and FF emissions are explained in Section 2.2.2.

We produced estimates for each year's posterior emission covariance error matrix using cost function gradient values from the

L-BFGS method that we employ to minimise the cost function (Nocedal, 1980), based on the method suggested by Bousserez et al. (2015). This iterative method estimates the inverse of the hessian (the second derivative) of the cost function, and does not include the off-diagonal elements of the posterior covariance matrix, so the posterior errors described in this manuscript are likely to be lower limits (Bousserez et al., 2015).

**2.2.2 Prior emissions and chemical sinks of CH4**

Prior emissions were taken from a range of widely available bottom-up models and inventories. Anthropogenic emissions were originally taken from the EDGAR v4.2 FT 2010 inventory (Olivier et al., 2012) and scaled as in McNorton et al. (2018) to apply an increasing global linear trend for the period after 2012. Biomass burning emissions were taken from GFEDv4.2 (van der Werf et al., 2017). The JULES model (Clark et al., 2011) was used to provide wetland fluxes, in a configuration described

in McNorton et al. (2016), using four separate carbon pools to drive methanogenesis. Rice emissions were taken from Yan et al., (2009) and are scaled by a factor of 0.75 as in Patra et al. (2011). Remaining small natural sources (termites, geological and oceanic emissions) were included as in Wilson et al. (2016). Sectoral emission maps are shown in Figure A2 whilst prior totals for each source type within South American regions are shown in Table 1. The prior global flux of CH$_4$ to the atmosphere rises from 549.5 Tg yr$^{-1}$ in 2009 to 564.0 Tg yr$^{-1}$ in 2017. The surface soil sink due to methanotrophs was from the Soil

Methanotrophy Model (MeMo, Murguia-Flores et al., 2018), repeating the 2009 flux every year, with a value of 33.9 Tg yr$^{-1}$. Landfill and fossil fuel emissions had temporal correlations imposed in the prior uncertainty matrix and made up the FF category, whilst the remaining emissions (NAT + AGR + BB) had no prior temporal correlations imposed. Atmospheric OH fields, based on those provided within the TransCom CH$_4$ study (Patra et al., 2011) were taken from Spivakovsky et al. (2000) and scaled downwards by 8% in accordance with Huijnen et al. (2010). The OH fields used here have previously been shown

to capture observed atmospheric lifetimes for CH$_4$ and methyl chloroform (CH$_3$CCl$_3$) in TOMCAT to within the observed uncertainty, although simulations of sulfur hexafluoride (SF$_6$) and other species show that the interhemispheric gradient in TOMCAT is slightly large compared to observations but within the bounds of other transport models (Patra et al., 2011; Wilson et al., 2014). The OH fields vary from month to month but do not vary between years. Montzka et al. (2011), Naus et al. (2020) and Patra et al. (2021) suggested that variability in annual OH mole fractions is small, but other recent research has suggested

the possibility of a declining trend in OH since 2004 (Rigby et al., 2017; Turner et al., 2017), although this trend had a high level of uncertainty. Other studies have found that the El Niño – Southern Oscillation (ENSO) has had a significant impact on OH variability in the troposphere in recent decades (e.g. Rowlinson et al., 2019; Anderson et al., 2020; Zhao et al., 2020), and

potentially an increasing trend in tropospheric OH during 1980 -2010 (Zhao et al., 2020). A trend in OH, or any year-to-year variability, was not included in our analysis, which will inform our conclusions, but for now we do not have enough evidence to include any potential variations. Stratospheric loss fields due to reactions with atomic chlorine (Cl) and excited oxygen atoms (O($^1$D)) varied on a monthly and annual basis and were taken from a previous full chemistry simulation from TOMCAT (Monks et al., 2017). The total simulated atmospheric $CH_4$ sink due to reaction with the OH radical in 2009 was 494.5 Tg, whilst the annual stratospheric $CH_4$ sink due to O($^1$D) and Cl was 19.5 Tg. $CH_4$ loss in the troposphere through reaction with Cl was not included in these simulations.

**2.2.3 Bottom-up model**

We also use a simple bottom-up (B-U) model to estimate wetland $CH_4$ emissions from meteorological and ecological input data, so that we can investigate the causes of variations in $CH_4$ emissions derived in the inversion. The B-U model, which is based on observed or modelled estimates of wetland fraction, heterotrophic respiration of carbon and surface temperature, is described fully in Appendix B. The model uses measurements of gravity anomalies made on the twin Gravity Recovery and Climate Experiment (GRACE) satellite mission as a proxy for variations in wetland fraction. The equation that our B-U model is based on is commonly used in other studies that estimate wetland fluxes of $CH_4$ (e.g. Clark et al., 2011; Melton et al., 2013; Bloom et al., 2017), but our application of the driving climate variables is fairly simple relative to these previous works. This method is sufficient for this work as the purpose of the B-U model is to investigate the possibility of reproducing the inversion results, and if they can be reproduced, to learn how and why the $CH_4$ wetland emissions change according to the input variables. If the inversion results aren't reproduced using the B-U wetland model, it could indicate that other sectors played a role in any observed variation.

**3 Results**

**3.1 Average distribution of emissions**

Average GOSAT $XCH_4$ over South America since 2009 show that $XCH_4$ column mole fractions were largest over the west of the continent, particularly in the northwest (Figure 2). Using the prior flux distribution in TOMCAT leads the model to underestimate $XCH_4$ in the northeast and far north of the continent and in the outflow into the Atlantic Ocean. Simulated $XCH_4$ is overestimated to the south and west of the continent. After assimilation, the largest model biases are removed, although there is a small positive bias in the interior of the continent, usually smaller than 5 ppb. The posterior (prior) mean model-satellite mismatch, weighted by the observation uncertainty, is 0.2 (-24.1) ppb globally, -5.4 (-40.0) ppb within South America and -4.1 (-66.5) ppb within the Amazon basin. The posterior residuals show no significant trend or seasonality within South America or within the basin (Figure A3).

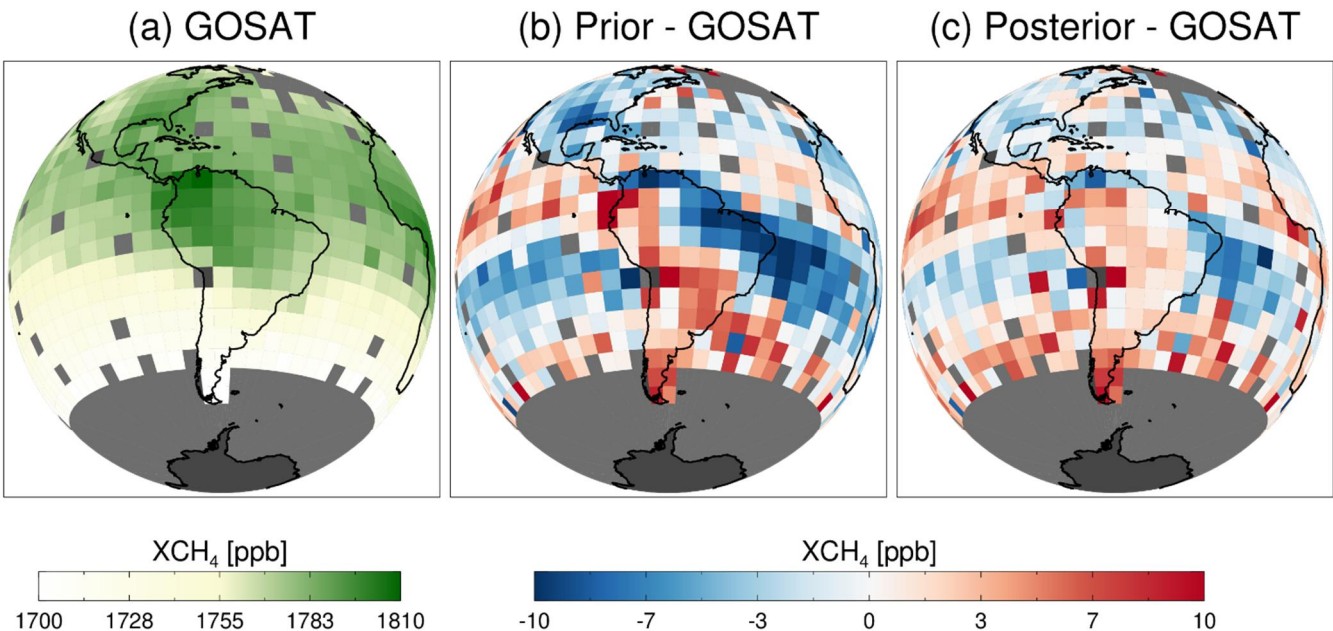

**Figure 2: (a) Mean GOSAT XCH$_4$ over South America and surrounding area for 2009 - 2018. Observations have been averaged onto the TOMCAT model grid as described in the text. Also shown is the mean difference between the model and satellite XCH$_4$ using (b) the prior emissions and (c) the posterior emissions for the same period.**

Figure 3 shows the 2009 – 2018 mean prior and posterior emission distributions of CH$_4$ emissions in South America. We display the mean over this entire period in order to show the consistent, long-term emission distribution. Whilst posterior uncertainty in particular grid cells can still be fairly large (Figure A4), regional changes are much less uncertain. Posterior South American emissions are significantly redistributed compared to the prior distribution due to changes in the NAT + AGR + BB emission sectors. Whilst the prior emissions are fairly homogeneous across much of the Brazilian Amazon, the posterior emissions are largest to the north-east of the continent and are reduced in the south and the north-west. Emission rates in the far north of the continent, potentially related to seasonal flooding in the basin of the Orinoco river in Venezuela, are also high in the posterior results.

The most significant feature of the posterior distribution is a region of high emission rates near the coastal basins around the mouth of the Amazon river itself (Fig. 3). There are large emissions from the region around the north-eastern states of Para, Maranhão and Tocantins. These areas contain the basins of many of the larger Amazon tributaries and a high density of wetland sources such as marshes, swamps and mangroves, according to the Sustainable Wetlands Adaptation and Mitigation Program (SWAMP) data from the Center for International Forestry Research (CIFOR) (Gumbricht et al., 2017). The prior flux distribution also highlights agricultural sources near this region in the EDGAR inventory (Fig. A2).

However, in our posterior results, the western Amazon and the Pantanal region in the south of Brazil do not display high emissions. Although the coarse resolution of the model grid boxes masks the signal from the relatively small Pantanal region in the prior emissions to some extent, it is still surprising that the posterior emissions have small methane fluxes from the southern regions of Brazil. As shown in Fig. 2, the model generally overestimates the $XCH_4$ in southern Brazil compared to GOSAT when using the prior emissions, so it is not surprising that emissions from that region were reduced in the inversion. The low emissions from a region where we expect significant methane release might mean that the model transport errors affect comparisons in this region, that the model-satellite bias included in the inversion (Eq. (1)) is inaccurate, or that the GOSAT sampling does not cover this region well (note the relatively small error reduction in this region, Fig. A4). The low emissions in the western Amazon are also a consistent feature of our results. The FF emissions do not change significantly in the inversion, although they are slightly decreased towards the southeast of Brazil, close to the large cities of São Paulo and Rio de Janeiro. The overall pattern of the posterior emissions displayed in Figure 3 is robust on a year-to-year basis, with the change relative to the prior in each individual year displaying very similar patterns to the multi-year mean (Figure A5).

**Table 1: Prior and posterior emissions of CH$_4$ for Brazil and other subregions of South America (2010 – 2017). Units are Tg yr$^{-1}$.**

| | Prior (Tg yr$^{-1}$) | | | | Posterior (Tg yr$^{-1}$) | | | |
|---|---|---|---|---|---|---|---|---|
| | 2010-2013 | | 2014-2017 | | 2010-2013 | | 2014-2017 | |
| | NAT + AGR + BB | FF | NAT + AGR + BB | FF | NAT + AGR + BB | FF | NAT + AGR + BB | FF |
| **Brazil** | 38.9 ± 11.7 | 10.6 ± 9.2 | 38.2 ± 11.4 | 10.6 ± 9.2 | 39.9 ± 5.3 | 9.9 ± 0.9 | 45.7 ± 5.1 | 9.9 ± 0.9 |
| **South America** | 59.9 ± 16.4 | 23.9 ± 16.2 | 58.5 ± 16.0 | 23.9 ± 16.2 | 62.7 ± 7.0 | 31.6 ± 1.7 | 68.9 ± 6.7 | 28.9 ± 1.8 |
| **West Brazilian Amazon** | 10.1 ± 5.4 | 0.3 ± 0.5 | 10.2 ± 5.5 | 0.3 ± 0.5 | 9.7 ± 2.9 | 0.3 ± 0.0 | 12.0 ± 2.8 | 0.3 ± 0.0 |
| **East Brazilian Amazon** | 13.4 ± 7.1 | 2.7 ± 3.8 | 12.9 ± 6.7 | 2.7 ± 3.8 | 20.0 ± 3.4 | 2.4 ± 0.3 | 24.3 ± 3.3 | 2.5 ± 0.3 |
| **Non-Amazon Brazil** | 15.4 ± 6.3 | 7.5 ± 8.4 | 15.1 ± 6.2 | 7.5 ± 8.4 | 10.2 ± 2.9 | 7.2 ± 0.8 | 9.3 ± 2.8 | 7.1 ± 0.9 |
| **Amazon basin** | 35.6 ± 12.4 | 4.1 ± 4.3 | 35.1 ± 12.2 | 4.1 ± 4.3 | 38.2 ± 5.3 | 3.5 ± 0.3 | 45.6 ± 5.2 | 3.7 ± 0.3 |

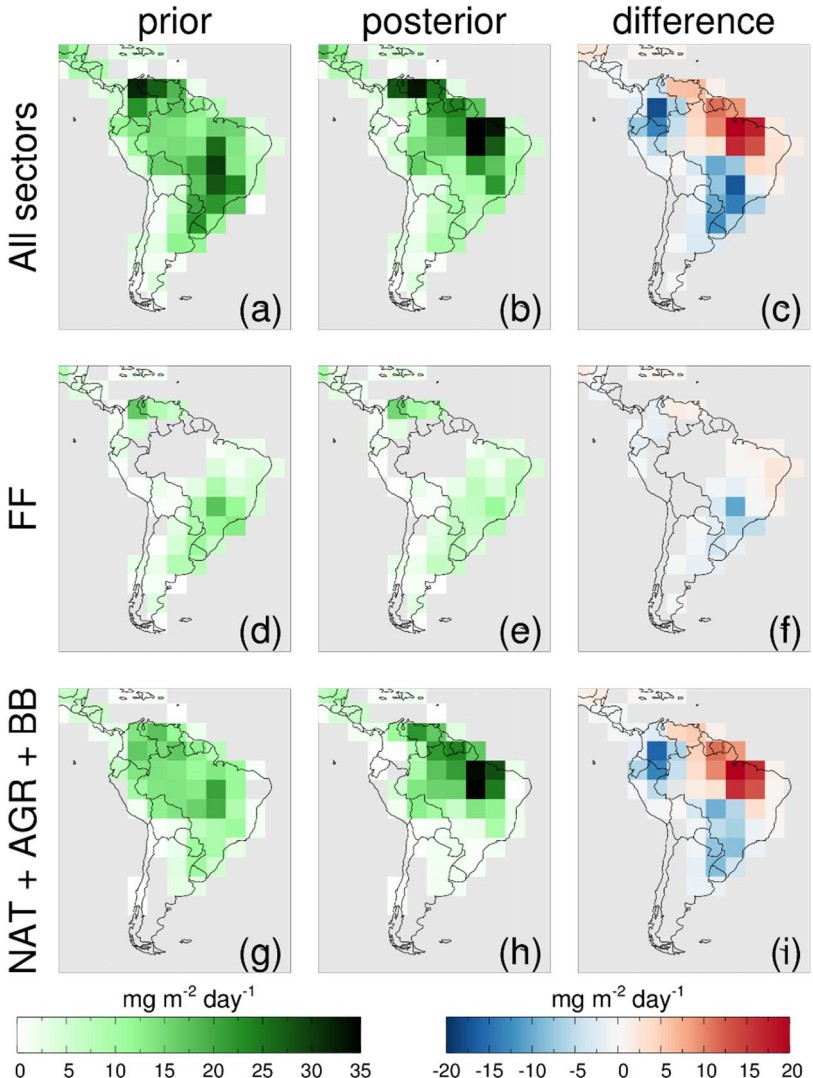

**Figure 3: Prior, posterior and (prior – posterior) mean gridded total South American CH₄ emissions (mg m⁻² day⁻¹) for the period 2009 – 2018 (a-c), and similar but for fossil fuel sources (d-f) and natural/agricultural/biomass burning sources (g-i) only.**

## 3.2 Temporal variations of CH₄ emissions

The annual total prior emissions in Brazil are nearly constant over time (Figure 4), with a mean value of $48.6 \pm 14.9$ Tg yr⁻¹ (uncertainty here from prior error covariance matrix). However, the posterior emissions show a positive trend, particularly from 2013 onwards. Globally, the posterior flux rises from 566.2 Tg yr⁻¹ in 2009 – 2013 to 594.0 Tg yr⁻¹ in 2014 – 2018 (Table 1), consistent with other studies (e.g. Saunois et al., 2020). In Brazil, the mean posterior annual emissions are $49.8 \pm 5.4$ Tg yr⁻¹ in the period 2009 – 2013, but rise to $55.6 \pm 5.2$ Tg yr⁻¹ in 2014 – 2018, with a mean value over the whole period of $52.7 \pm 5.3$ Tg yr⁻¹. The uncertainty stated for these figures represents the overall mean annual posterior uncertainty for Brazil derived

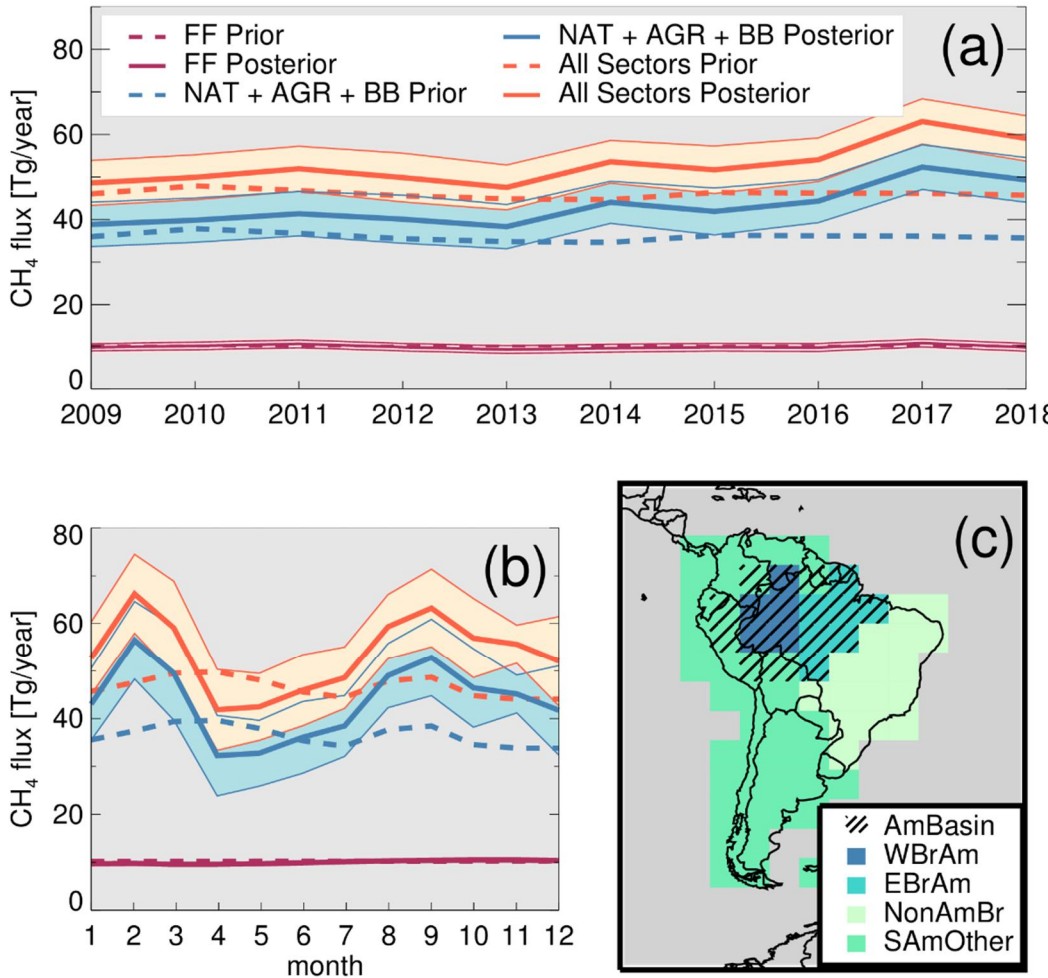

**Figure 4: (a) Total annual Brazilian prior and posterior emissions (Tg(CH₄)/yr). Shaded areas show posterior uncertainties as derived in the inversion. (b) Monthly mean prior and posterior Brazilian CH4 emissions (Tg(CH₄)/yr, 2009 – 2018). Shaded areas show standard deviation for each month. (c) Regions of South America discussed in the text. Hatched area (AmBasin) represents the Amazon basin across all countries, whilst the shaded areas show Brazilian and non-Brazilian regions.**

in the inversion for each 4-year period. We report the mean annual uncertainty as we assume that posterior uncertainty for each year is strongly correlated with that in other years. Reporting the mean value implicitly assumes a correlation of 1 between the years' uncertainties; in reality the correlation is likely smaller than 1. Our total Brazilian mean flux is within the range found by Saunois et al. (2020), and agrees well with the findings of Janardanan et al. (2020). There is a significant positive trend over the whole time period (2010 – 2018) of $1.37 \pm 0.69$ Tg yr$^{-2}$ ($p < 0.05$), driven by the NAT + AGR + BB emissions category, although the distribution actually resembles a step-change in 2014.

350

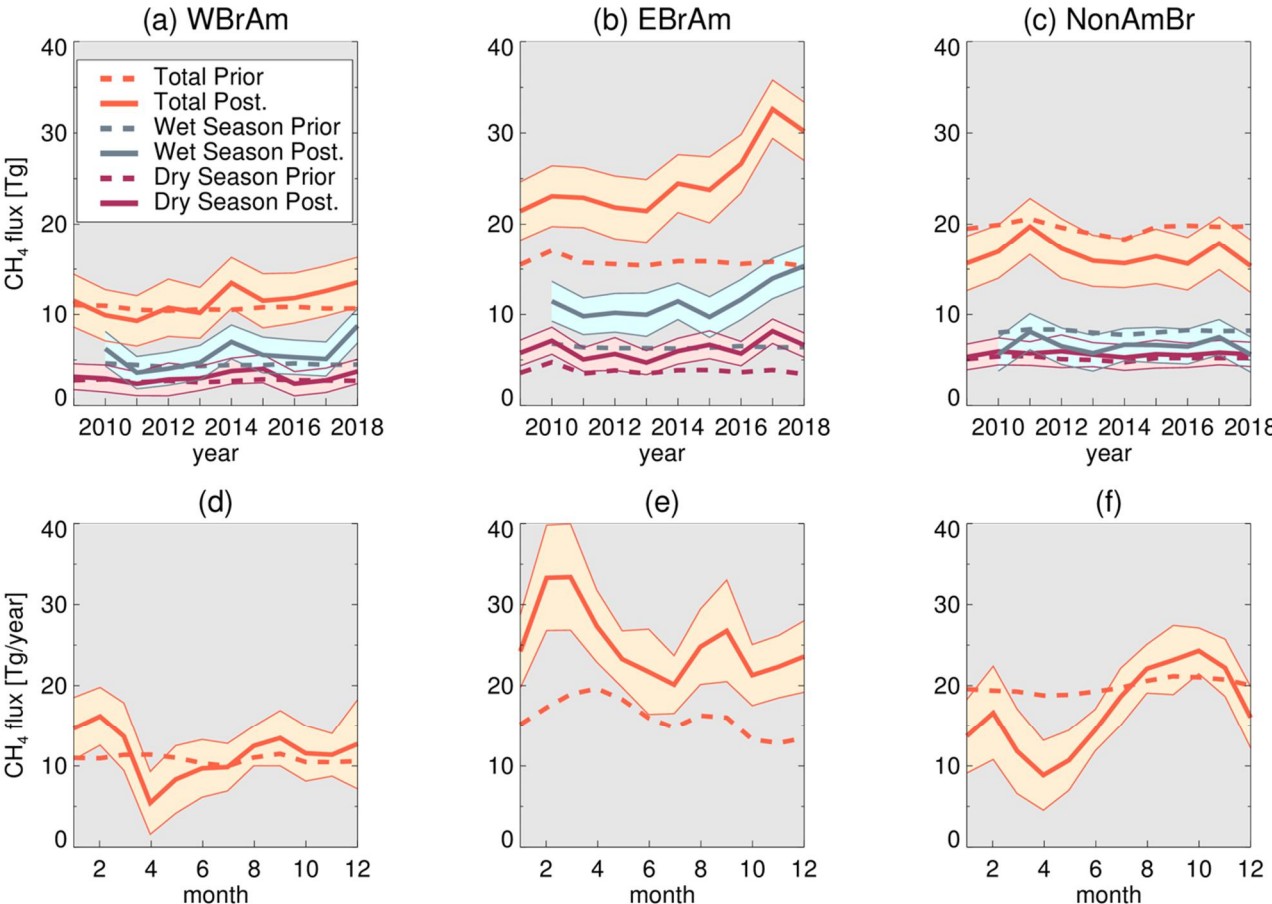

**Figure 5: (a-c)** Total annual (red lines) prior and posterior emissions of CH$_4$ (Tg yr$^{-1}$) in three Brazilian subregions; the western Brazilian Amazon (WBrAm), the eastern Brazilain Amazon (EBrAm) and non-Amazon Brazil (NonAmBr). Prior and posterior emissions during the wet season (December – March, gray lines) and the dry season (August – October, maroon lines) are also shown. Shading represents the posterior uncretainties for each region derived in the inversion. **(d-f)** Monthly mean prior and posterior emissions for the period 2009 – 2018 (Tg yr$^{-1}$) for the three sub-regions. Shading shows the standard deviation of the monthly means.

Posterior emissions in Brazil peak in February and September (Figure 4b), representing the wet-season and dry-season, most
likely due to contributions from the local seasonal cycles of wetland emissions and biomass burning emissions, depending on the location. The peak monthly emission rate of $66.2 \pm 8.2$ Tg yr$^{-1}$ is in February, before lower rates of emission during April to July. This February peak corresponds to a peak in precipitation across the basin (from the Global Precipitation Climatology Project (GPCP) v2.3 combined precipitation dataset (Adler et al., 2018)), but precedes the peak in gravity anomaly – representative of soil water depth – captured by the GRACE satellite (Figure A6). Emissions in August and September are
almost as large as those during the peak of the wet season. Again, almost all of this seasonal variation comes from the NAT + AGR + BB emission category.

Within the entire Amazon region, emissions grew from $41.7 \pm 5.3$ Tg yr$^{-1}$ in 2010 – 2013 to $49.3 \pm 5.1$ Tg yr$^{-1}$ in 2014 – 2017. Emissions are largest in the eastern Brazilian Amazon (EBrAm, Figure 5), and are significantly larger than suggested by the prior emissions, particularly in the most recent years. The increase in emissions over the period is also largest there, rising from $22.4 \pm 3.4$ Tg yr$^{-1}$ in 2010 – 2013 to $26.8 \pm 3.3$ Tg yr$^{-1}$ in 2014 – 2017 (trend: 1.06 Tg yr$^{-2}$, p<0.01). Emissions also increase from $10.0 \pm 2.9$ Tg yr$^{-1}$ to $12.3 \pm 2.8$ Tg yr$^{-1}$ between these two periods in the western Brazilian Amazon (WBrAm). However, in the non-Amazon region of Brazil (NonAmBr), emissions decrease slightly over these years (from $17.5 \pm 3.0$ Tg yr$^{-1}$ to $16.4 \pm 2.9$ Tg yr$^{-1}$). Trends in WBrAm and NonAmBr are not significant for p<0.01. The Amazon regions of Brazil display the two-peak seasonal cycle of CH$_4$ emissions, although this is much more pronounced in the east. This is at least partly due to the significant effect of biomass burning within the arc of deforestation in the southeast of the basin that usually occurs during these months. Emissions are largest in NonAmBr during the dry season, possibly due to fires or agricultural activity.

We also show total emissions for each subregion during the wet season (December – March) and the dry season (August – October). These periods were defined using the GPCP precipitation data, as periods when the average monthly precipitation during 2009 – 2018 within the basin was more than 7 mm day$^{-1}$ and less than 3 mm day$^{-1}$, respectively. In both WBrAm and EBrAm, the trends for the 2009 – 2018 period are largest in the wet season. This suggests that trends in wetland and floodplain emissions could be responsible for the rising CH$_4$ emissions, in line with reports of intensifying flood extremes in the area in recent decades (Barichivich et al., 2018). However, there are other potential explanations. These include escalating biomass burning emissions during the wet season (Silva Junior et al., 2019) and an intensification of agricultural emissions in these regions, as seen in version 5.0 of the EDGAR anthropogenic flux inventory in Brazil (Crippa et al., 2020), or some combination of factors. Unfortunately, our results cannot be used to say more about which sectors are responsible for the increasing flux.

### 3.3 Comparison to independent observations

Observations of CH$_4$ made during flights within the basin between 2010 and 2018 were used to independently check the performance of the prior and posterior emission distributions in the model (Figure 6, Table 2). For the observations made at altitudes higher than 3km, which represents the free troposphere above the Amazon, the mean bias (MB) between model using the posterior emissions and the observations is significantly reduced at all locations, compared to that produced when using the prior flux in the model. The correlation between the model and the observations also increases at all locations when using the posterior flux rather than the prior. The absolute value of the model – observation bias is reduced to below 6 ppb at all sites. However, the posterior model MB against observations made in the boundary layer, at altitudes below 1.5 km, is higher than the prior model MB at three sites. At the western sites, RBA and TAB/TEF, the MB in the model increases by approximately 15 ppb, although the correlation improves, particularly at TAB/TEF. At ALF, the correlation decreases slightly, and the MB increases by a large amount (31 ppb). Finally, at SAN, the performance improves significantly by both measures, with the MB being reduced from -47.8 ppb to -15.2 ppb. There are no significant trends (at 95% level) in the model – aircraft

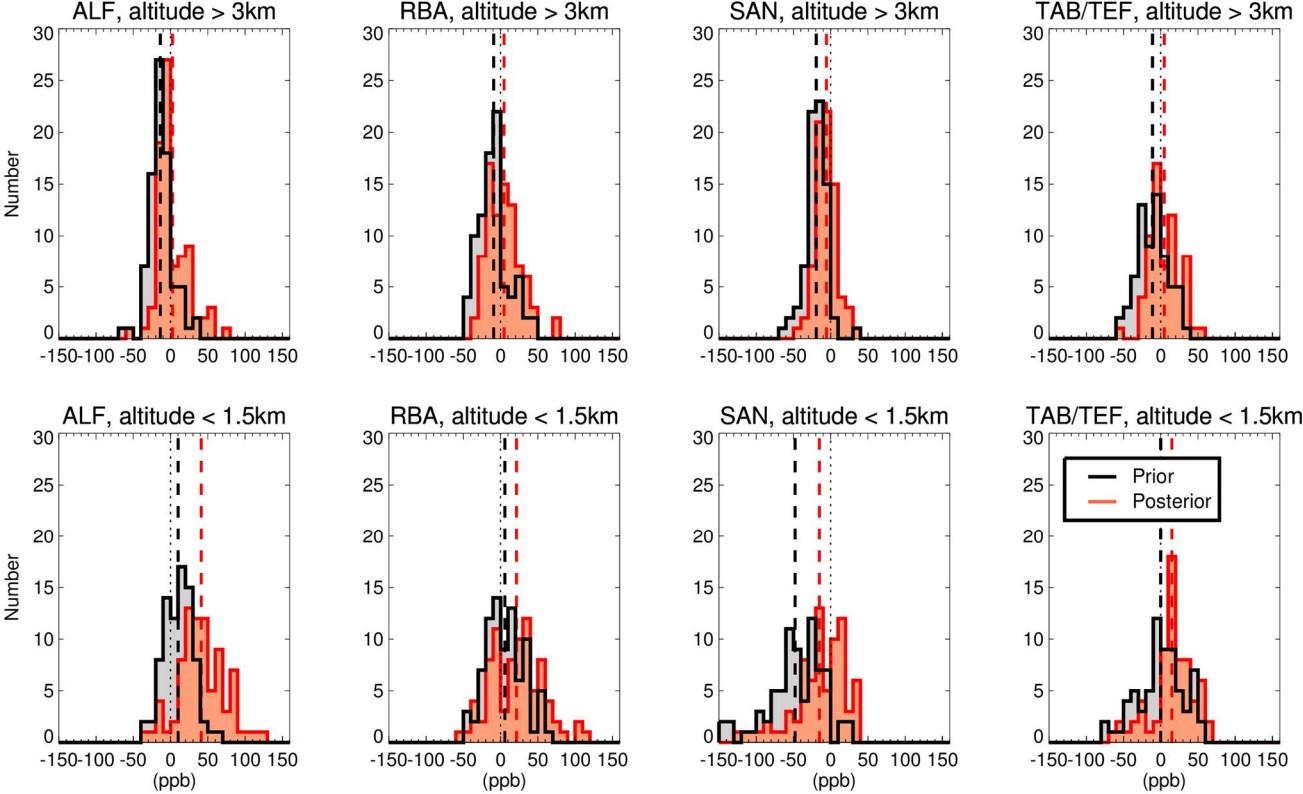

**Figure 6: Histogram plots showing prior (black) and posterior (red) [model – observation] differences at the four Amazon flight locations, 2010 -2018. Measurements were taken at Alta Floresta (ALF), Rio Branco (RBA), Santarém (SAN) and Tabatinga and Tefé (TAB/TEF). Model output has been interpolated to observations locations and altitudes, before both were averaged into monthly means and into altitude bins of 3km and above (a-d) and 1.5km and below (e-h). Dotted vertical lines show the zero line, whilst dashed vertical lines show prior and posterior mean model – observation bias.**

residual biases in 2010 – 2017, except at TAB/TEF below 1.5km. This site has a posterior residual bias trend of +2.1 ppb/year, but this may have been caused by the change in the flight location halfway through the study period.

The improved performance at SAN is significant, as the high mole fractions of $CH_4$ sampled at this location relative to expectations given its location situated close to the eastern coast have been previously noted (Miller et al., 2007; Basso et al., 2016; Wilson et al., 2016). The prior model therefore has a large negative bias at SAN, particularly near the surface. The posterior distribution of emissions, with a region of high emission rates to the south and east of the basin, substantially reduces the model – observation difference at SAN. The model still underestimates methane mole fractions at this site even after the improvement, however, which might be due to bias which remains in the posterior flux estimate, possibly due to the allocated prior uncertainty in this region being too small, or model representation uncertainty. The fact that ALF is also located near these significant emissions leads to degradation in the model performance within the boundary layer, which was previously better at ALF than at SAN. The capability of assimilation of GOSAT $XCH_4$ to improve performance at both of these locations

might have been reduced due to the relatively coarse model grid. Webb et al. (2016) found that comparisons between the flight-based observations and a previous version of the GOSAT XCH$_4$ used in this study showed that the GOSAT values were larger than equivalents estimated using the flight data at ALF, but that the discrepancy was much smaller at SAN. This being the case, it is not surprising that the model in which the GOSAT data has been assimilated has difficulties in matching the flight observations at both locations at once. Since we assimilated XCH$_4$ from GOSAT, which is mostly representative of the troposphere, it is expected that the model performance is improved at all locations when compared to observations made at the higher altitudes. This also indicates good model representation of inflow of CH$_4$ to the basin from elsewhere. However, the fact that the posterior comparisons are generally degraded close to the surface, apart from at SAN, mean that the local sources close to these sites might be overestimated at this model resolution, that there are errors in the model's representation of vertical mixing, or that there remains a positive bias in the assimilated retrievals from GOSAT in this region. Generally, however, the temporal variation and MB in the model is much improved after the assimilation of GOSAT XCH$_4$.

**Table 2: Prior and posterior bias (ppb) and correlation between TOMCAT and Amazon flight observations (2010 – 2018). The better value for bias and correlation for each site and altitude is highlighted in bold.**

|  | Prior mean bias (ppb) | Posterior mean bias (ppb) | Prior correlation | Posterior correlation |
|---|---|---|---|---|
| ALF, >3km | -13.7 | **2.6** | 0.71 | **0.75** |
| SAN, >3km | -19.4 | **-5.7** | 0.79 | **0.88** |
| TAB/TEF, >3km | -11.1 | **4.7** | 0.67 | **0.81** |
| RBA, >3km | -9.4 | **4.6** | 0.70 | **0.80** |
| ALF, <1.5km | **10.2** | 41.2 | **0.70** | 0.67 |
| SAN, <1.5km | -47.8 | **-15.2** | 0.32 | **0.49** |
| TAB/TEF, <1.5km | **0.0** | 15.0 | 0.48 | **0.65** |
| RBA, <1.5km | **5.7** | 21.4 | 0.54 | **0.56** |

### 3.4 Bottom-up model results

The inversion suggests that CH$_4$ emissions have been significantly increasing from eastern Amazon regions throughout the 2010s, but is not able to determine the source sectors responsible for this rise. The largest increases over time occur during the wet season (Figure 5), when wetland emissions dominate the atmospheric signal, so it is possible that changes to these emissions led to the increase. Emissions from seasonal floodplains and wetlands are sensitive to temperature, precipitation (which affects wetland area) and carbon availability in the soil (Bloom et al., 2017), so we examined these factors to see how they varied during the previous decade and whether wetlands could have been responsible for increasing wet season CH$_4$ emission in the basin.

The mean surface temperature within the Amazon basin increased throughout the period 2009 - 2018 (Figure 7), while there was no significant trend in precipitation (not shown) or gravity anomaly. Estimating the trends of these factors is significantly affected by one anomalously dry and hot period, running from late 2015 to mid-2016. These record-breaking conditions were caused by the 2015/16 El Niño, and were largely confined to the east of the basin (Jiménez-Muñoz et al., 2016). A previous extreme event during this study period, in the dry season of 2010, displayed a similar geographical distribution but was easily surpassed by the scale of the 2016 drought (Lewis et al., 2011; Jiménez-Muñoz et al., 2016). One other event that stands out is the prolonged flooded period running through the wet season of 2013/14, during which rainfall in the south-west of the basin was up to twice as much as usual (Espinoza et al., 2014). This flooded period did not coincide with a significant ENSO period but was likely caused by warm conditions in the Subtropical South Atlantic.

Figure 7 also shows the wet season mean anomalies for each year for the surface temperature, gravity anomaly and modelled heterotrophic respiration. Wet season temperatures were high in 2010 and in 2015, 2016 and 2018. The water table was at its highest in 2012, 2014 and 2015. Finally, heterotrophic respiration was strongest in 2010, 2013 and 2014, but very low in 2015 and 2016. There was no heterotrophic respiration model data available for 2017, so we used a climatology value for that year. We felt that this was justified since the temperature and water table depths also had only very small anomalies during that season. As might be expected, the temperature and gravity anomalies in the wet season were strongly negatively correlated (r=-0.66), due to coincidence of hot and dry conditions.

The temperature trend in the Amazon was positive throughout almost the entire basin (Figure 8a), particularly in the far west and south east. The trend in the wetland fraction (Figure 8b) was more heterogeneous, with positive trends in the west contrasting with strong negative trends across the east of the basin. Both sets of trends are strongly impacted by the hot, dry conditions in 2015/16.

The geographical distribution of the NAT + AGR + BB wet season $CH_4$ emission trend produced by the inversion (Figure 8c) is positive across the north west and south east of the basin, with a similar distribution to the temperature trends. The positive emission trends in the north west are collocated with an area of increasing wetland fraction. However, the regions to the east and south with strong positive emission trends are located where wetland fraction had been decreasing as temperatures increased. This suggests that, if wetlands were responsible for increased wet season flux, the methanogenesis must have been more sensitive to the increasing temperature than to the decrease in wetland fraction or in heterotrophic respiration (not shown).

We ran the B-U model multiple times, varying the temperature response and the GRACE anomaly scaling variables in order to produce a range of likely values for $CH_4$ flux from the basin. We also optimised B-U model parameters, as described in Section 2.2.3, in order to best reproduce the INVICAT results using the B-U model (Figures 8d and 8e). The B-U model combines the three driving variables, but the strong anti-correlation between the temperature and wetland fractions mean that this model does not produce strong variations in emissions, since the two tend to cancel out. Using the optimised B-U model

produces weak positive emission trends in the west of the basin, and weak negative trends elsewhere, giving no significant trend overall (p = 0.36). The optimised values are included in Appendix B. The standard deviation of the wet season emissions in the B-U model is 1.7 Tg yr$^{-1}$, compared to 2.4 Tg yr$^{-1}$ in the inversion results. The mean posterior error in in the inversion results (2.9 Tg yr$^{-1}$) is relatively large compared to the standard deviation, however, meaning that the B-U values almost always remain within the posterior inversion uncertainty. The exception to this is the wet season of 2014, when the inversion results

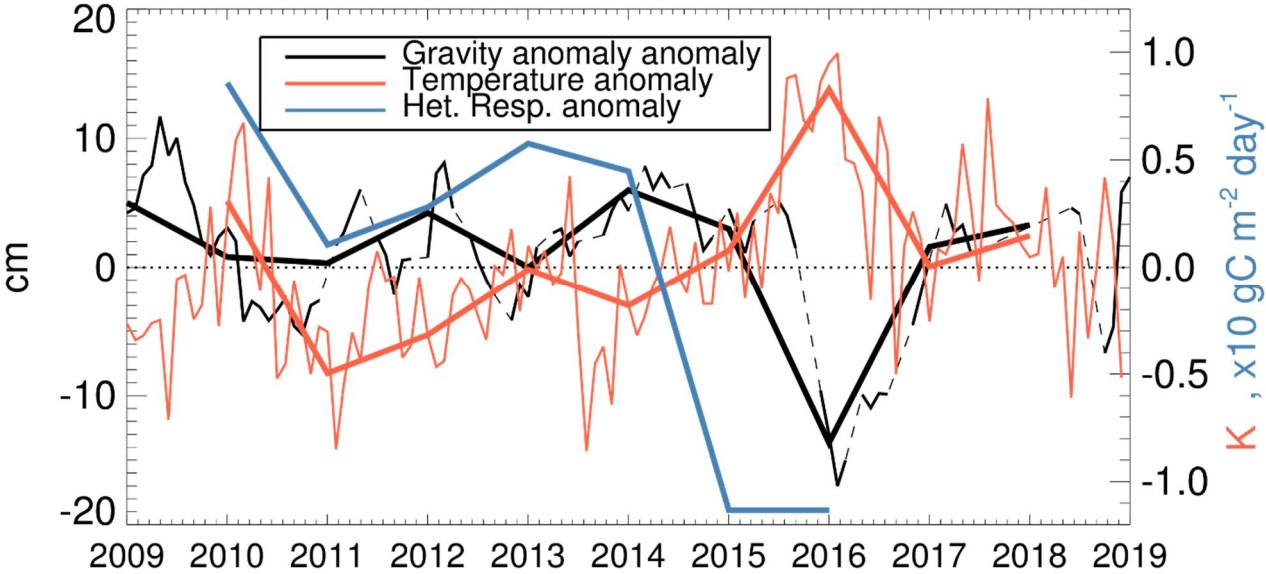

**Figure 7: Anomalies of gravity anomaly (cm, black, left axis), surface temperature (K, red, right axis), and heterotrophic respiration (x10 gC m-2 day-1, blue, right axis) for the period 2009 – 2018 within the Amazon basin. Monthly mean anomalies are shown as thin lines, whilst wet season (December – March) averages are shown as thick lines. Interpolated values for gravity anomalies are shown as dashed lines.**

produce larger emissions than in any other year (20.1 ± 2.7 Tg) and this feature is not reproduced by the B-U model. As discussed, the wet season of 2014 was subject to extreme precipitation and widespread flooding in the basin (Espinoza et al., 2014), and the GRACE gravity anomalies are large throughout this period (Figure 7), whilst heterotrophic respiration was high and temperatures were relatively cool (although warmer than in 2011 and 2012). Despite these conditions which seem favourable to $CH_4$ emission, the B-U model does not produce emissions significantly larger than any other year. The

discrepancy between the inversion and B-U model results is discussed further in Section 4. Figure 8e also shows the wet season emissions within the basin from the Full Ensemble (FE) of the WetCHARTS emission dataset (Bloom et al., 2017),

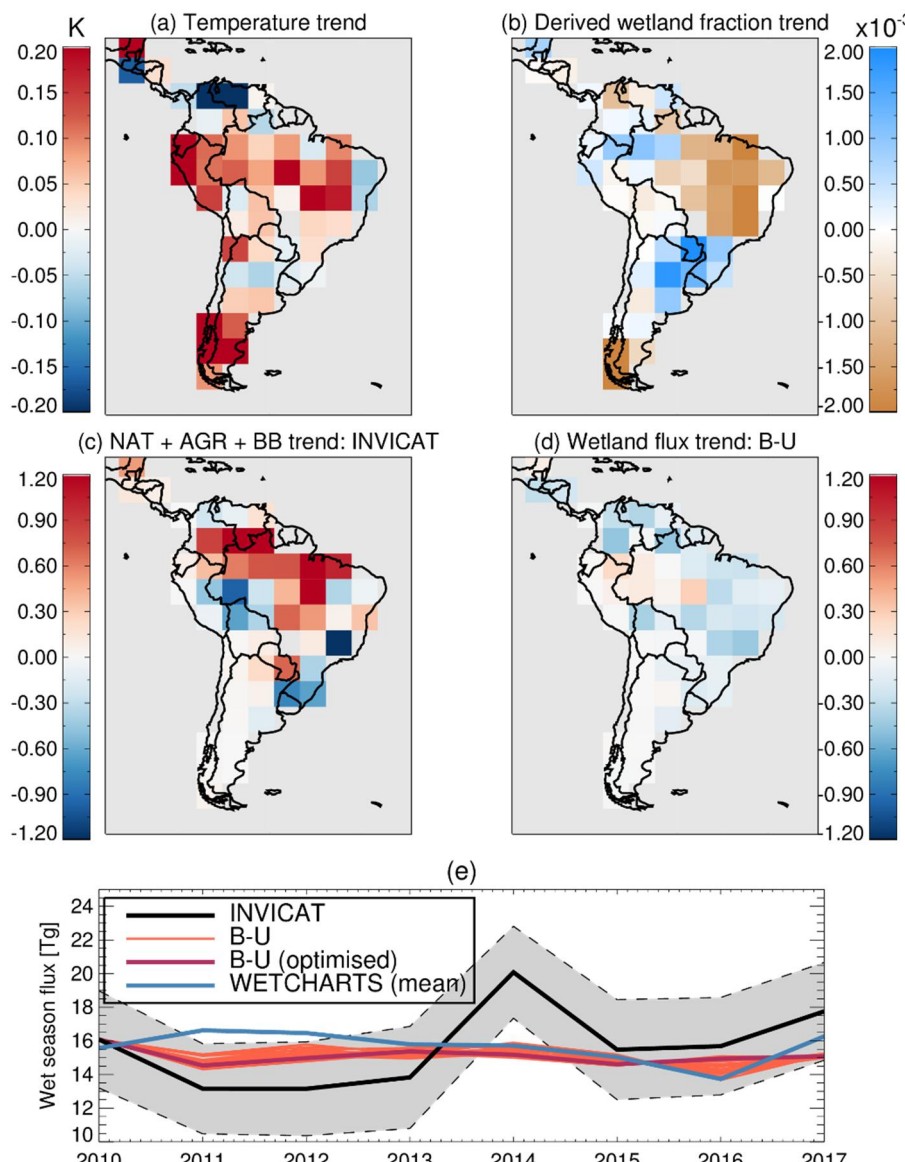

**Figure 8: Average wet season trends for the period 2010 – 2017 for (a) temperature in K year⁻¹; (b) wetland fraction in 10⁻³ year⁻¹; (c) NAT + AGR + BB CH₄ surface flux in mg m⁻² day⁻¹ year⁻¹ from GOSAT inversion; and (d) Optimised bottom-up (B-U) model surface flux of CH₄ in mg m⁻² day⁻¹ year⁻¹. (e) Total Amazon basin wet season CH₄ emissions in Tg (2010 – 2017) from GOSAT inversion (black line, with grey shading representing posterior uncertainty). Red lines show ensemble of B-U model simulations, and maroon line is the optimised B-U model. Blue line shows the mean of the WetCHARTS Full Ensemble wet season flux.**

which use a similar method to estimate wetland emissions that used in our B-U model. These emissions also show a negative trend over the period 2010 – 2017 (-0.17 Tg yr⁻¹), and the variation is again small (0.93 Tg yr⁻¹ standard deviation). They display no significant change in emissions in the wet season of 2013/14. The implications of the discrepancy between the inversion and the B-U model are discussed in Section 4.


**4 Discussion**

We derive emissions of $CH_4$ in Brazil for the period 2010 – 2018 of $52.7 \pm 5.3$ Tg yr$^{-1}$, split into two periods during which mean Brazilian emissions were $49.8 \pm 5.4$ in 2010 – 2013 and $55.6 \pm 5.2$ in 2014 – 2017, an increase of $5.8 \pm 5.2$ Tg yr$^{-1}$. This increase was found to be entirely due to the NAT + AGR + BB emissions within the Amazon region, In Amazonia, emissions
grew from $41.7 \pm 5.3$ Tg yr$^{-1}$ to $49.3 \pm 5.1$ Tg yr$^{-1}$ over the same two periods.

This increase between the two periods is very similar to that found by Tunnicliffe et al. (2020), although the total emissions found in our study are larger than their finding of $33.3 \pm 3.7$ Tg yr$^{-1}$. They removed a model – satellite bias of $22 \pm 8$ ppb from the GOSAT observations used in their study, which is much larger than our bias of $3 – 9$ ppb removed from $XCH_4$ over the Amazon. This treatment of bias, coupled with differences in model transport, could explain the different emissions derived.
The positive biases in our posterior $CH_4$ relative to aircraft observations within the boundary layer also suggests that our emissions could be overestimated. However, we note the absence of significant trends in our posterior model minus aircraft residuals between 2010 – 2017. Our posterior total emissions agree well with the findings of Janardanan et al., (2020) who derived Brazilian emissions of $56.2 \pm 10$ Tg yr$^{-1}$ for the period 2011 – 2017, although temporal variation of this value was not discussed in that study. Yin et al., (2020) did not report total emissions, but found a rise in Amazonian emissions of $4.2 \pm 1.2$
Tg yr$^{-1}$ over 2010 – 2017, along with small increases in eastern Brazil. Our estimate of total flux from Amazonia agrees well with that of Pangala et al. (2017), who derived a total of $42.7 \pm 5.6$ Tg yr$^{-1}$ for 2010 – 2013. A group of 22 inverse model experiments presented by Saunois et al., (2020) produced a range of $47.3 – 78.2$ Tg yr$^{-1}$ for Brazilian emissions between 2008 and 2017, although one of those results used the TOMCAT forward model to represent the atmospheric transport, so is not fully independent from our results. Our findings here are within the range of these models, albeit towards the lower end. The
majority of those top-down studies used either the same GOSAT and surface observation data used in our study, or some variation of it. The fact that the derived emissions using similar observation data can vary so much highlights the inherent uncertainties still remaining in top-down studies of $CH_4$ emissions, with differences in model transport, chemistry representation, inversion methodology, bias correction and error assumptions all contributing to differences in results.

The increase in emissions that we derive from 2014 onwards coincides with a faster rate of increase in the observed surface
mole fraction of $CH_4$ (Nisbet et al., 2019). Unfortunately, the extent that the increase in observed mole fractions in the atmosphere is driven by increasing Amazon emissions is difficult to constrain without more extensive knowledge of the atmospheric chemical loss of $CH_4$. Our global inversion, using repeating OH values each year, indicates that the increase of $5.8 \pm 5.2$ Tg yr$^{-1}$ from Amazon emissions is responsible for $24 \pm 18\%$ of the global total increase in emissions between 2010-13 and 2014-17, which was $24.1 \pm 15.0$ Tg yr$^{-1}$.


The Amazon emissions derived in this study for 2010 and 2011 ($41.6 \pm 5.3$ Tg yr$^{-1}$) are a little above the higher limit of those found in our previous study using the flight observations only (31.6 – 41.1 Tg yr$^{-1}$, Wilson et al., 2016). This indicates that using the vertical profile data only to calculate basin-wide emission totals may lead to a small underestimation of the total compared to using satellite data. This discrepancy is supported by the positive bias seen in this study within the boundary layer at most of the sites when comparing the posterior model output to the in situ flight observations. However, the emission totals are fairly similar across the different methodologies, with the caveat that the same transport model was used for both findings.

The posterior fluxes lead to an improvement compared to the prior in the mean bias and correlation at all four independent sampling locations when only observations made above the boundary layer are considered. However, the posterior comparison to observations made close to the surface are inferior to the prior comparison at three of the locations. It seems that improving the performance compared to the GOSAT data throughout the troposphere is at the expense of diminishing performance at the surface. There could, therefore, be transport errors in the inverse model, possibly in the boundary layer transport. It is possible also that the relatively coarse resolution of the inversion leads to poorer comparisons to the boundary layer observation. Finally, as stated by Webb et al. (2016), comparisons between the flight observations and GOSAT at the Alta Floresta (ALF) site, which displays the worst posterior performance in the model, are also not as strong as at other locations. Despite the increased posterior bias in the boundary layer at three of the sites, the improved performance at Santarém suggests that the significant emissions close to the mouth of the Amazon derived by the inversion are potentially a realistic feature, consistent with the previous in situ data-based flux estimates of Miller et al. (2007) and Basso et al. (2016). However, the degradation in performance at Alta Floresta, also in the east of the basin, suggests that the strong emissions do not extend as far south as in our model posterior. We will in the future produce inversions at higher resolution to investigate this feature further. Due to computational constraints, we could not carry out inversions for the entire GOSAT period using a higher horizontal resolution than the one chosen for our inverse model, but to examine the sensitivity of our results to the model resolution, we ran an inversion for 2010 at 2.8° horizontal grid resolution (Figure A7, Appendix C), finding that our results were robust at both resolutions.

Our derived positive trends are largest during the wet season within the east Amazon, indicating that increasing flux from wetland sources could be responsible for the increase in total emissions. However, attempting to reproduce these trends, and the interannual variations, using a B-U model was largely unsuccessful. Although the B-U model mainly stayed within the uncertainty derived in the inversion, it was unable to capture a large increase in emissions in the wet season of 2014. This indicates either that the variation produced in INVICAT was overstated, that there were errors in the B-U model set-up and input data, or that wetlands were not the main driving factor for increasing wet season emissions from the basin. It has been suggested that there have recently been unusually large biomass burning $CH_4$ flux in the region outside of the dry season (e.g. Silva Junior et al. 2019), or that agricultural emissions from enteric fermentation and manure were increasing at a significant pace (Crippa et al., 2020), neither of which would be captured in our B-U model .

Potential errors within the inverse model are likely due to one of five factors. The model transport, sink distribution and variation, error covariance matrices, satellite retrieval uncertainty and method of comparing the model and satellite can all affect the posterior results. Regarding the use of repeating OH values for each year of the inversion, however, it should be noted that Tunnicliffe et al., (2020) used a regional model in which the chemical sink of $CH_4$ was not a factor, and found similar levels of interannual variably to those produced here.

The performance of the B-U model compared to the inverse model suggests conflicting hypotheses. The positive trend in emissions produced in INIVCAT was concurrent with increasing temperatures across much of the Amazon. This indicates that the temperature response of wetland emissions in the region might be high. However, the fact that the B-U model was unable to produce significantly larger emissions during the 2014 wet season, as were produced by the inversion, despite large wetland fraction and heterotrophic respiration at the time, indicates that the wetland fraction response might also be high, and potentially non-linear. Comparing the results from the B-U model for 2012 and 2014 is instructive, as 2014 had higher heterotrophic respiration and temperature, and a similar (but slightly higher) mean wetland fraction. However, the B-U emission totals for these two years were very similar. Although the observed mean gravity anomalies were similar, they were characterised differently, with prolonged positive anomalies throughout 2013/14, but a short and intense positive anomaly during the end of the 2012 wet season. This suggests that emissions could be a function of the period of time for which the soil is saturated. It should be noted that Tunnicliffe et al., (2020) also derived large $CH_4$ fluxes during this wet season, but they were allocated to anthropogenic sources rather than wetlands using their methodology, likely due to differences in the transport model and sector allocation method. Increased complexity in the B-U model and examination of correlations between inversion-based fluxes and potential wetland flux drivers are both necessary for future comparisons, but for now it is not possible to determine the cause of the trend in $CH_4$ emissions in the Amazon basin.

## 5 Conclusions

Our global inversion of $CH_4$ emissions using satellite data and surface observations allowed us to quantify changes in South American emissions over the period 2009 – 2018. We found that emissions increased during this period, particularly during the wet season of December - March. Total Brazilian emissions rose from $49.8 \pm 5.4$ Tg yr$^{-1}$ in 2010 – 2013 to $55.6 \pm 5.2$ Tg yr$^{-1}$ in 2014 – 2017, whilst natural emissions from the Amazon basin (from all countries), an area of 6.9 million km$^2$ on this model grid, rose from $38.2 \pm 5.3$ Tg yr$^{-1}$ to $45.6 \pm 5.2$ Tg yr$^{-1}$. We show that there was substantial emission from the south and east of the basin throughout this period, and that the positive trends were largest in the eastern Brazilian Amazon. We derive particularly large emissions during the 2013/14 wet season, a period during which there was widespread flooding. It is significant that our inversions show improved performance at Santarém due to the large emissions in the east of the basin, similar to previous aircraft-based studies (Miller et al., 2007; Basso et al., 2016). Indeed, based on the remaining negative

model-observation bias at that location, it is possible that $CH_4$ emissions affecting that location could be even larger. However, it appears that the Alta Floresta site is overly affected by these large emissions in our analysis, indicating that the southerly extent of the large emissions might be too great.


However, attempting to reproduce these trends in a simple bottom-up model was unsuccessful, mainly due to strong anti-correlations between the wetland fraction and the temperature within the basin leading to little variation in annual wet season emissions. This suggests that the complexity of the model must be increased in order to fully represent the relationship between carbon availability, wetland fraction and soil temperature. Our B-U model, and another model (Bloom et al., 2017), suggest a

negative trend in emissions from driving conditions, but this is at odds with our inversion findings and those of others. This suggests that temperature has a strong role to play in wetland emissions of $CH_4$ in the Amazon region, since this has also had an increasing trend over the past decade. It is also important to consider the role of wetland variability, however. For the inverse model, the contribution of how sinks of $CH_4$ in the atmosphere might have varied should also be considered.

The results of our inversion are in agreement with previous studies (e.g. Janardanan et al., 2020), and within the range provided by Saunois et al. (2020). However, our posterior emissions from Brazil are significantly larger than those produced by Tunnicliffe et al., (2020) using a similar observational data set, showing the importance of model transport and bias correction in inversion results.

Our results show that the Amazon basin was responsible for $24 \pm 18\%$ of the total global increase in $CH_4$ emissions during the last decade, and it could contribute further in future due to its sensitivity to increasing temperature. Our study shows the benefit of using satellite $CH_4$ data to inform on emissions of $CH_4$, particularly in poorly sampled tropical regions, along with the benefits of long-term satellite missions to produce large-scale, consistent datasets. As the satellites and models improve, we can further refine our estimates of emissions from the important and changing role of South American ecosystems on global

methane variability.


## Appendix A

**Table A1: Locations and time periods covered by surface flask samples used in inversions, provided by the National Oceanic and Atmospheric Administration's Global Monitoring Laboratory.**

| Station code | Longitude, Latitude (°) | Time Period | Station code | Longitude, Latitude (°) | Time Period |
|---|---|---|---|---|---|
| ABP | 321.8E, 12.8S | 2009 – 2010 | LLN | 120.9E, 23.5N | 2009 – 2018 |
| ALT | 297.5E, 82.5N | 2009 – 2018 | LMP | 12.6E, 35.5N | 2009 – 2018 |
| AMY | 126.3E, 36.5N | 2013 – 2018 | MEX | 262.7E, 19.0N | 2009 – 2018 |
| ASC | 345.6E, 8.0S | 2009 – 2018 | MHD | 350.1E, 53.3N | 2009 – 2018 |
| ASK | 5.6E, 23.3N | 2009 – 2018 | MID | 182.6E, 28.2N | 2009 – 2018 |
| AZR | 332.6E, 38.8N | 2009 – 2018 | MKN | 37.3E, 0.0S | 2009 – 2011 |
| BAL | 17.0E, 55.4N | 2009 – 2011 | MLO | 204.4E, 19.5N | 2009 – 2018 |
| BHD | 174.9E, 41.4S | 2009 – 2018 | NAT | 324.8E, 5.8S | 2010 – 2018 |
| BKT | 100.3E, 0.2S | 2009 – 2018 | NMB | 15.0E, 23.6S | 2009 – 2018 |
| BME | 295.3E, 32.4N | 2009 – 2010 | NWR | 254.4, 40.0N | 2009 – 2018 |
| BMW | 295.1E, 32.3N | 2009 – 2018 | OXK | 11.8E, 50.0N | 2009 – 2018 |
| BRW | 203.4E, 71.3N | 2009 – 2018 | PAL | 24.1E, 68.0N | 2009 – 2018 |
| BSC | 28.7E, 44.2N | 2009 – 2011 | PSA | 296.0E, 65.0S | 2009 – 2018 |
| CBA | 197.3E, 55.2N | 2009 – 2018 | PTA | 236.3E, 39.0N | 2009 – 2011 |
| CGO | 144.7E, 55.2N | 2009 – 2018 | RPB | 300.6E, 13.2N | 2009 – 2018 |
| CHR | 202.8E, 1.7N | 2009 – 2018 | SDZ | 117.1E, 40.7N | 2009 – 2015 |
| CIB | 355.1E, 41.8N | 2009 – 2018 | SEY | 55.5E, 4.7S | 2009 – 2018 |
| CPT | 18.5E, 34.4S | 2010 – 2018 | SHM | 174.1E, 52.7N | 2009 – 2018 |
| CRZ | 51.9E, 46.4S | 2009 – 2018 | SMO | 189.4E, 14.3S | 2009 – 2018 |
| DRP | 296.3E, 59.0S | 2009 – 2018 | STM | 2.0E, 66.0N | 2009 |
| DSI | 116.7E, 20.7N | 2010 – 2018 | SUM | 321.6E, 72.6N | 2009 – 2018 |
| EIC | 250.5E, 27.2S | 2009 – 2018 | SYO | 39.6E, 69.0S | 2009 – 2018 |
| GMI | 144.7E, 13.4N | 2009 – 2018 | TAC | 1.1E, 52.5N | 2014 – 2015 |
| HBA | 333.8E, 75.6S | 2009 – 2018 | TAP | 126.1E, 36.7N | 2009 – 2018 |
| HPB | 11.0E, 47.8N | 2009 – 2018 | THD | 235.8E, 41.1N | 2009 – 2017 |
| HSU | 235.3E, 41.0N | 2009 – 2017 | TIK | 128.9E, 71.6N | 2011 – 2018 |
| HUN | 16.7E, 47.0N | 2009 – 2018 | USH | 291.7E, 54.9S | 2009 – 2018 |
| ICE | 339.7E, 63.4N | 2009 – 2018 | UTA | 246.3E, 39.9N | 2009 – 2018 |
| IZO | 343.5E, 28.3N | 2009 – 2018 | UUM | 111.1E, 44.5N | 2009 – 2018 |
| KEY | 279.8E, 25.7N | 2009 – 2018 | WIS | 35.1E, 30.0N | 2009 – 2018 |
| KUM | 205.0E, 19.7N | 2009 – 2018 | WLG | 100.9E, 36.3N | 2009 – 2018 |
| KZD | 76.9E, 44.1N | 2009 | ZEP | 11.9E, 78.9N | 2009 – 2018 |
| KZM | 77.9E, 43.2N | 2009 | | | |

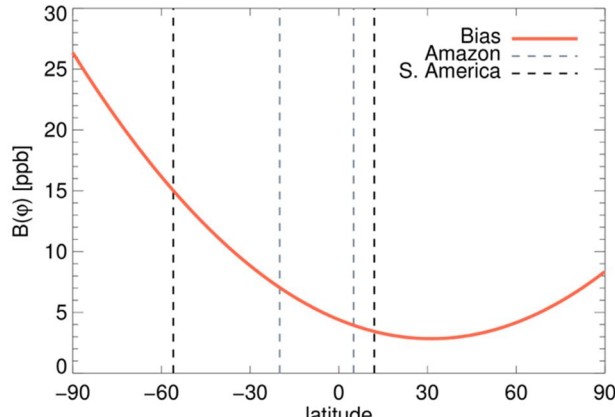

**Figure A1: Model – satellite bias function $B(\varphi)$ (red line) varying with latitude, $\varphi$. This value is added to the simulated XCH$_4$ to enable comparison with the GOSAT XCH$_4$. See main text for details. Vertical grey and black dotted lines show the latitudinal extent of the Amazon basin and South America, respectively.**

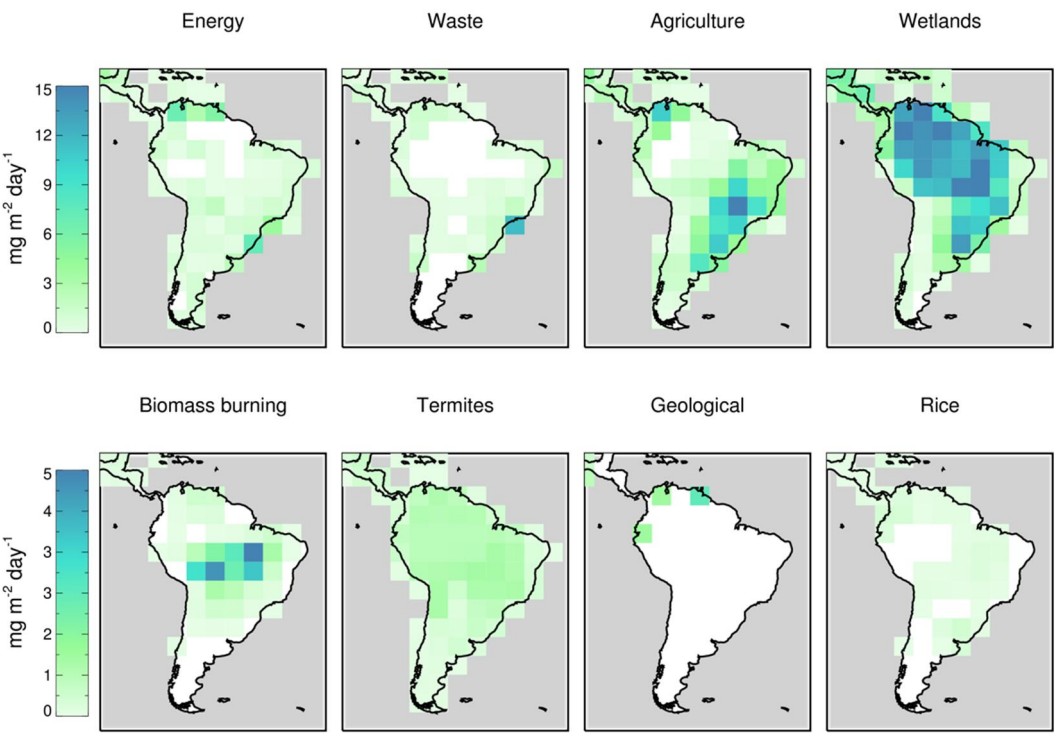

**Figure A2: *A priori* gridded sectoral South American CH$_4$ emissions (mg m$^{-2}$ day$^{-1}$) for 2010. Note different colour scales for top and bottom rows. Description of sectoral emissions are in Section 2.2.2 of the main text. Here, the 'Energy' sector refers to the energy industry, oil and gas production and energy for buildings and transportation; 'Waste' refers to solid waste disposal and waste water; 'Agriculture' refers to enteric fermentation and manure management.**


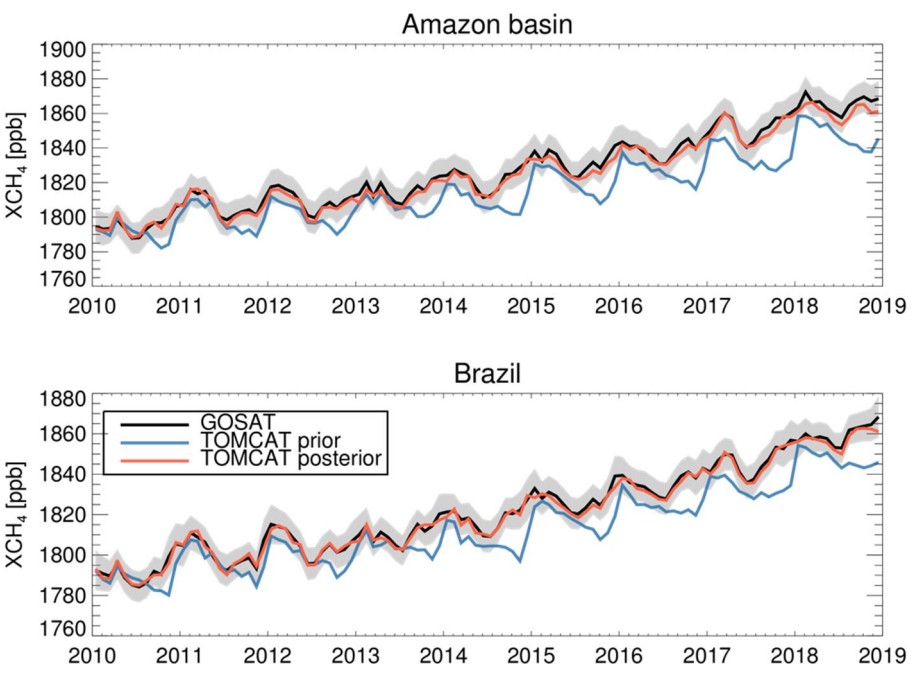

**Figure A3: Error-weighted monthly mean GOSAT XCH₄ (black lines) over the Amazon basin (top) and Brazil (bottom). Also shown are the simulated mean prior (blue) and posterior (red) XCH₄ with GOSAT averaging kernels applied. Grey shading shows the monthly mean observation uncertainty within each region. It should be noted here that since simulations for each year were initiated using values from the previous year's posterior mean output for both the prior and the posterior, the gradient of the blue line is artificially higher than it would be for a free-running simulation using the prior emissions only.**



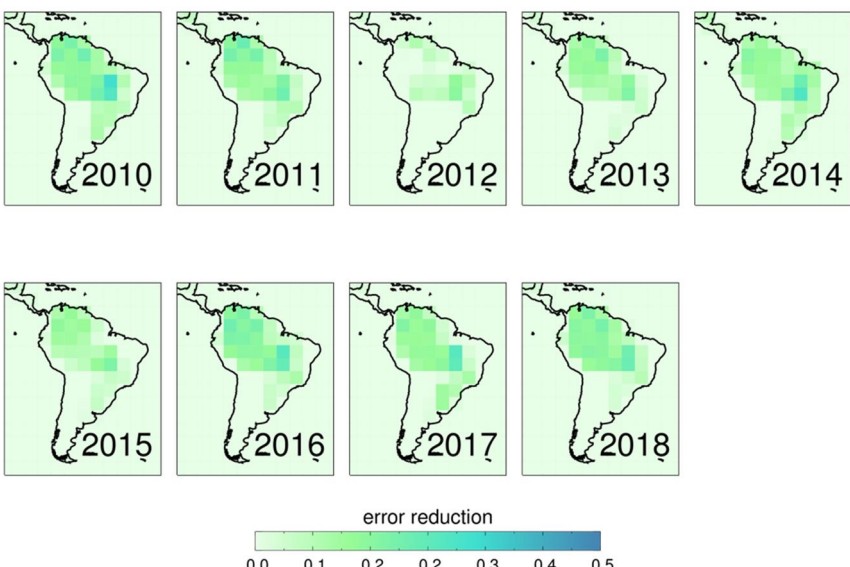

**Figure A4: Mean annual South American posterior error reduction after assimilation of GOSAT XCH4 and surface flask observations.** Error reduction is defined as 1.-($\sigma$_a/$\sigma$_b ), where $\sigma$_a is the derived standard deviation of the a posteriori grid cell flux uncertainty and $\sigma_b$ is the allocated *a priori* standard deviation of the grid cell flux uncertainty.

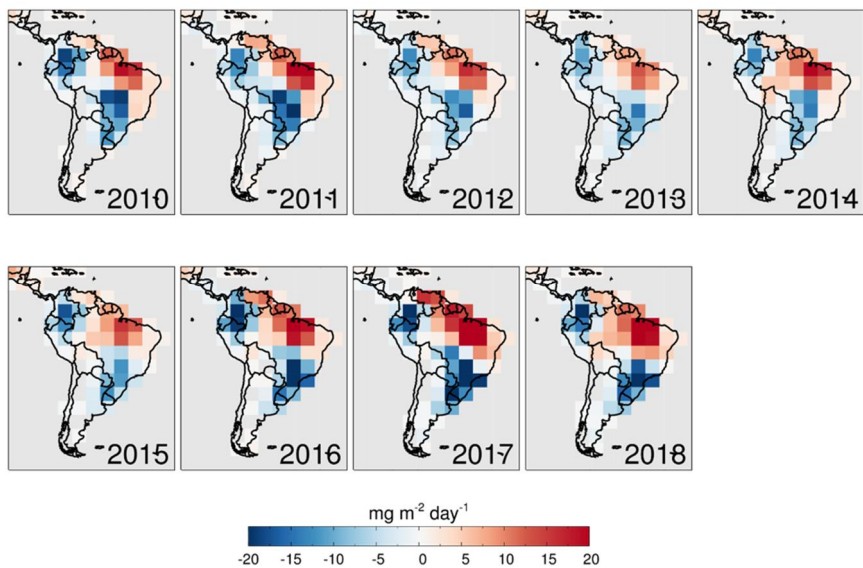

**Figure A5: Annual mean (posterior – prior) gridded total South American CH4 emissions (mg m$^{-2}$ day$^{-1}$) for each year covering the period 2010 – 2018.**

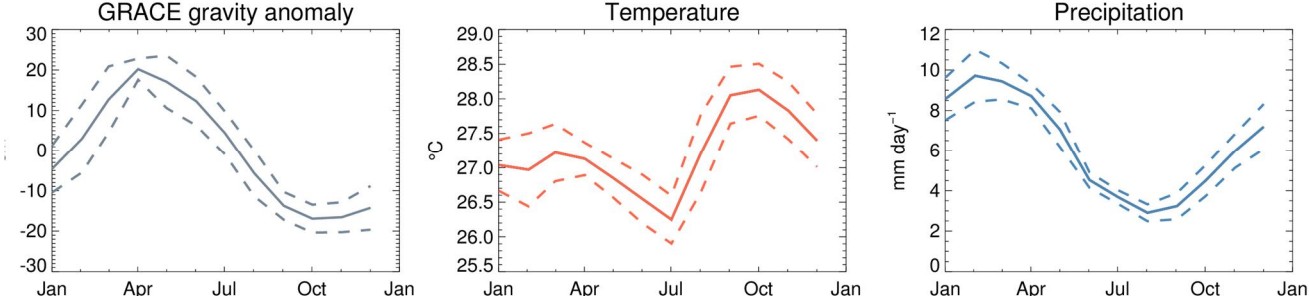

**Figure A6: Mean seasonal cycle of GRACE gravity anomaly (left, cm), temperature (centre, °C) and precipitation (right, mm day$^{-1}$) within the Amazon basin for 2010 – 2018. Dashed lines show one standard deviation from the mean values. Temperature is taken from the NOAA/NCEP Global Historical Climatology Network v2 and the Climate Anomaly Monitoring System GHCN Gridded v2, whilst precipitation is from the Global Precipitation Climatology Project v2.3 combined precipitation dataset.**

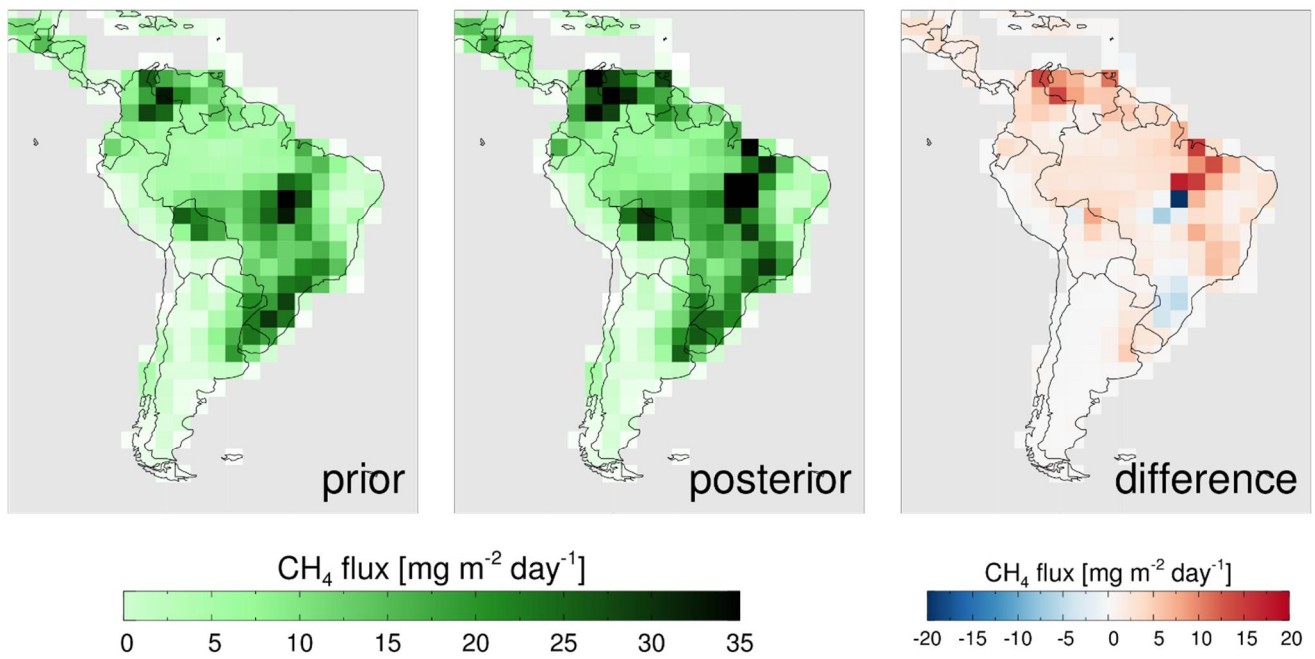

**Figure A7: Prior (left) and posterior (centre) emissions of CH$_4$ (mg m$^{-2}$ day$^{-1}$) for 2010 from an inversion carried out on the 2.8° degree model grid. (Right) Posterior – prior emissions.**

## Appendix B: Bottom-up wetland flux model

Our bottom-up (B-U) model calculates wetland $CH_4$ emissions, in which the $CH_4$ emissions in a grid cell, $x$, at time, $t$, are dependent on climatological factors as follows:

$$F(t, x) = s\, A(t, x)\, R(t, x)\, q_{10}^{\frac{T(t,x)}{10}} ,$$ (2)

where $F(t, x)$ is the flux of $CH_4$ in molecules $cm^{-2}\ s^{-1}$, $A(t, x)$ is the wetland fraction, $R(t, x)$ is the heterotrophic respiration of carbon per unit area, $T(t, x)$ is the surface temperature in °C, and $q_{10}$ is the relative $CH_4$:C ratio of respiration for a 10°C change in temperature. Finally, $s$ is a scaling factor. We use monthly mean values for each element of Eq. (2) and interpolate all parameters to the TOMCAT model grid for comparison with the inversion results.

We take $R$ from the CASA-GFED v4.1 model (Randerson et al., 2015), which runs up to 2016, and gridded 2m temperature from the NOAA/NCEP Global Historical Climatology Network version 2 and the Climate Anomaly Monitoring System GHCN Gridded V2 data provided by the NOAA Physical Sciences Laboratory (https://psl.noaa.gov/, Fan and Dool, 2008). We estimate $A$ using a combination of two products. We take a climatology of wetland fraction $w(x)$ from the JULES land surface model version that was used to produce the prior emissions used in the inversion (McNorton et al., 2016). We then use measurements of gravity anomalies made on the twin GRACE satellite mission, $G(t, x)$ as a proxy for variations in the soil moisture, as in (e.g.) Bloom et al. (2010) and Gloor et al. (2018). We then apply scaling factors $a_1$ and $a_2$ to give wetland fraction as follows:

$$A(t, x) = a_1 w(x) + a_2 G(t, x) ,$$ (3)

This makes the assumption that anomalies in the gravity anomaly $G(t, x)$ are linearly related to wetland fraction anomalies, which may not be the case. The distributions and variations of the GRACE gravity anomalies and surface temperature are discussed in Section 4. We create an ensemble of B-U estimates for $F$, letting the scaling factors $a_1$ and $a_2$ and the temperature response function $q_{10}$ vary within reasonable limits, and varying $s$ appropriately so that each member gives the same mean total emissions over 2010 – 2017, equal to the mean posterior value produced by the inversion. We are interested only in the variations in time and space produced by the B-U model, rather than the absolute value. We let $q_{10}$ vary between 1 and 3, based on experimental bounds and previous bottom-up studies of methane emissions (Yvon-Durocher et al., 2014; Bloom et al., 2017), we let $a_1$ vary between 0.8 and 1.2, and we let $a_2$ vary in such a way that the overall wetland fraction does not vary by more than 20%, depending on the value of $a_1$. Since there is no data for 2017 given for the heterotrophic respiration, we use a climatology made up from the preceding seven years applied to that year. We also create an 'optimised' B-U model, in which we use a curve-fitting procedure (based on a gradient expansion algorithm) to choose values of $s$, $a_1$, $a_2$ and $q_{10}$ which

best fit, in least-squares terms, the results from the inversion for the monthly and spatial mean values over the whole Amazon, for all months within the wet season over 2010 - 2017. The optimised value of $q_{10}$ was 2.47, which is within the range of plausible values discussed in Section 2.2.3, whilst the optimised values of $a_1$ and $a_2$ were 0.73 and 0.0015, respectively. For comparison to the B-U model, we consider only the posterior inversion NAT + AGR + BB emissions within the Amazon basin during the wet season, which we assume to be almost entirely from wetlands (Fig. A2). We therefore also only consider the

B-U model output during the wet season.

**Appendix C: Increased resolution inversion**

We ran an inversion for 2010 at 2.8° horizontal grid resolution (Figure A7) averaging the GOSAT XCH$_4$ onto this model grid.

We did not split the results into different source sectors, instead deriving total CH$_4$ surface flux. Otherwise, the model set-up was identical to the 5.6° inversions of the main study. Many of the features of the posterior solution are identical to those of the coarser grid, with higher emissions from the region to the south and east of the Amazon river, and a decrease in emissions from the south of Brazil, near the densely populated cities. However, there is no decrease in emissions to the west of the Amazon basin, as consistently seen when using the coarser model grid. Total derived emissions for Brazil and for the Amazon

basin are similar when using the 2.8° and the 5.6° grids, however. We derive total posterior emissions for Brazil in 2010 of 49.9 Tg yr$^{-1}$ using the coarser grid, and 51.4 Tg yr$^{-1}$ using the finer grid.

*Author contributions.* C.W., M.P.C. and M.G. designed the methodology and wrote the manuscript. C.W. performed the

analysis. R.J.P and H.B. provided the GOSAT data. J.M. and S.A.M. provided TOMCAT model input. L.V.G., J.B.M. and L.S.B provided the Amazon flight data. All authors contributed with analysis and text.

*Code and data availability.* University of Leicester GOSAT Proxy XCH$_4$ data can be accessed via the Copernicus Climate Data Store (https://cds.climate.copernicus.eu/cdsapp#!/dataset/satellite-methane?tab=overview ) or by contacting Rob Parker.

Prior and posterior mean South American CH$_4$ fluxes on the TOMCAT model grid (Wilson et al., 2021) are available from the data archive of the Centre for Environmental Data Analysis (CEDA, https://www.ceda.ac.uk). The data DOI is 10.5285/88224a922439441fa6644b4564dcd90c.

*Competing interests.* We declare no conflicts of interest.


**Acknowledgements**

CW, MPC, RJP and HB are funded via the UK National Centre for Earth Observation (NE/R016518/1 and NE/N018079/1). The work at Leeds was also supported by the NERC MOYA project (NE/N015657/1). We thank the Japanese Aerospace Exploration Agency, National Institute for Environmental Studies, and the Ministry of Environment for the GOSAT data and their continuous support as part of the Joint Research Agreement. This research used the ALICE High Performance Computing Facility at the University of Leicester for the GOSAT retrievals. LVG, LSB, JBM are funded via the following grants: FAPESP (16/02018-2, 11/51841-0, 08/58120-3, 18/14006-4), NASA grants (11-CMS11-0025, NRMJ1000-17-00431), ERC Horizon 2020 (649087) and 7FP EU (283080). We thank Wuhu Feng for support for the TOMCAT model and Ross Bannister for advice regarding error covariance.

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
