# Peer review of "Large and increasing methane emissions from Eastern Amazonia derived from satellite data, 2010 - 2018"

_Atmospheric Chemistry and Physics, 2020_

## Referee Comment (RC1) · Luke Western (Referee) · 30 Nov 2020

**Review of the article: "Large and increasing methane emissions from Eastern Amazonia derived from satellite data, 2010–2018" by Wilson et al.**

The manuscripts presents estimates of methane emissions, focused on Brazil and the Amazon Basin, using a top-down inversion approach, which is validated against independent measurements, and compared to a bottom up model. The manuscript is thorough and detailed, which will undoubtedly be informative for future studies. The conclusions are supported by the findings of the study, which are suitably appraised. As such, I hope to see its eventual publication in ACP. There is however, a major error

in the methodology – as it is stated in the paper – which may or may not have a large impact on the results. This must be addressed before the manuscript can be considered acceptable for publication. In addition, below are a number of suggestions for revisions to improve the manuscript, followed by technical comments.

- 1. Line 150-151: "For accurate comparison between the retrieved XCH4 and those simulated by the model, the GOSAT averaging kernels were averaged similarly to the XCH4 and applied to the model vertical profiles." This approach is not mathematically sound and therefore, as an absolute minimum, it must be checked that it has a negligible effect. Otherwise it could lead, quite likely, to an underestimation of the modelled XCH4 and thus overestimate emissions. The reason for the error here is as follows. If we simplify the maths to just two variables, and let X be the concentration at each of the *i* the model levels, and A be the diagonal of the averaging kernel, and Y be the XCH4, then we can say that  $Y = \sum_i a_i x_i$ . If we take the mean of each observation *j* of the total *n* observations in a grid cell then we want  $Y_{grid} = \frac{1}{n} \sum_j Y_j$ , which is equal to  $\frac{1}{n} \sum_j \sum_i a_{ij} x_{ij}$ . As there is only one X per grid cell in the model, we can simplify this to  $\frac{1}{n} \sum_j \sum_i a_{ij} x_i^{grid}$ . From the text, it states that instead the average of the averaging kernel was applied to the modelled profile, so  $Y_{grid} = \sum_i \frac{1}{n} (\sum_j a_{ij}) x_i^{grid} = \frac{1}{n} \sum_i x_i^{grid} \sum_j a_{ij}$ , which is needs more clarification.
- 2. Line 197: Although each has completed 40 iterations, how do you check that the optimisation routine has converged within 40 iterations?
- 3. The paper is thorough and detailed, although as a matter of opinion it is in places quite arduous to read. I suggest making use of supplementary material and moving some of the analysis here. For example, at line 533: This section should be moved to the supplement, and referred to in line 531. The application of the B-U model as a whole and its discussion would be better placed in the supplement to improve the conciseness of the manuscript. This could also apply
to the validation against the site and aircraft data – refer to the outcomes in the main text but the details can be moved to a supplement. This should get across the key points of the paper, i.e. the emissions and their sources, better to the reader.

4. Code and data availability: It would be much more beneficial to the community if the results of this work (posterior emissions estimates in space and time) were publicly available. E.g. placing the spatial maps for the mean posterior emissions for Brazil and the posterior emissions estimates for Brazil each year in a public repository in e.g. netcdf format. I see that this has also been suggested by the topical editor, but I would like to reiterate its importance.

**Technical comments:**

Throughout: Units for ACP should be expressed in exponential form, i.e. Tg  $yr^{-1}$  and not Tg/yr.

Throughout: Be consistent with Tg(CH4)/yr and Tg/yr (e.g. line 81). I would recommend stating explicitly in its first occurrence that Tg/yr refers to Tg(CH4) and from thereon just writing Tg/yr.

Throughout: There is an error in the nomenclature used throughout the manuscript. The manuscript often refers to the prior/posterior mole fraction when referring to a single parametric estimate; the prior/posterior are distributions of values. A better usage is a priori and a posteriori, or more explicit, the prior/posterior mean.

Line 74: Basin is capitalised (not consistent with earlier use of basin).

Line 75: This sentence is confusing: it reads as though fires contribute to the number of wetland sources. Consider splitting this sentence as e.g. "as well as a number of other wetland sources in S America, emissions from....also contribute to methane emissions."
Line 77: Overlap in which sense? The emissions processes? In space? In stochastic error?

Line 79: Consider "variability" rather than "variance", so as not to confuse with statistical variance.

Line 79: 'Earlier estimates' is inexact. Is this when the research was carried out, or the emissions from the year(s) in question? Specify the time period that you are discussing.

Line 142: '...and found that the two agreed within their respective errors'. This sentence is meaningless without a description of the probability content in which these dataset agree (e.g. the 1 s.d. uncertain regions for both datasets overlap).

Eq(1): Why are the numbers in bold, as well as the brackets? Please remove.

Line 170: Specify that it is the inset of Figure 1.

Line 171: 'until 2014' should not be in parenthesis.

Line 172: Until when? Present?

Line 178: Space needed between 500 and m

Line 190: It is worth mentioning the species here, considering it is only 4 citations.

Line 194: Are these 5.6 degree square grids? Horizontal is vague.

Line 196: Use the latex command \citep[ERA-I][]{dee\_reference} here.

Line 215: They are given 250

Line 222: What was the functional family of the spatial correlation imposed? The most defensible choice here is a Matérn covariance structure (see Stein 2012, Interpolation of Spatial Data: Some Theory for Kriging), although it seems that this is not the case here.

Line 223: Again, probability content of this uncertainty needs defining.
Line 226: L-BFGS is a general minimisation routine, not a method to explicitly derive a covariance matrix in the context of uncertainty. This needs rewording.

Line 228: The 'cost function' is introduced for the first time here, and needs expanding upon for readers unfamiliar with the method.

Line 229: This should be the other way round – the lack of off-diagonals would give smaller emissions uncertainty than expected when including off-diagonals i.e.  $var(\sum_{i}(x_i)) = \sum_{i}(var(x_i)) + \sum_{i \neq j} cov(x_i, x_j)$ .

Line 237: Again, use \citep[MeMo][]{ref}.

Eq 2.: Brackets are bold, when they shouldn't be.

(Paragraph starting 274: This paragraph reads well and is explicit. Ideally much more of the paper should read like this.)

Line 301: 'The posterior error-weighted mean residual model-satellite mismatch' needs defining. It's unclear what this is.

Figure 2: Consider earlier comment about the use of the terms prior/posterior. Note that the red-blue colour bar is not colour blind friendly.

Line 310: Is this the mean of the prior/posterior means?

Line 318: This should say the spatial distribution of the posterior mean.

Line 345: What correlation do you assume when you make the assumption that they are highly correlated between years?

Figure 5: These colours are very difficult to differentiate. I suggest revising the shadings/colours in the figures.

Line 458: The curve fitting programme (note 'programme' if using British English) needs elaborating on. What is the programme? What curve does it fit?

Line 462: If talking about 'no significant trend', the (statistical) significance of this trend
must be given.

Line 474: Does 'here' refer to Section 4? If so, this should be stated as such: "Section 4 also shows..."

Line 551: Use  $\citep[e.g.][]{ref}$

Line 585: An extra closing bracket is present.

Line 586: Consider 'substantial' instead of significant.

Line 598: Which other models besides Bloom et al.?

Appendices Figures: Consider making these colour blind friendly.

ACPD

---

## Referee Comment (RC2) · Luke Western (Referee) · 14 Dec 2020

I would like to apologise for an error on my part in the review of this manuscript.

My comment (1) beginning: "Line 150-151: "For accurate comparison between the retrieved XCH4 and those simulated by the model, the GOSAT averaging kernels were averaged similarly to the XCH4 and applied to the model vertical profiles." This approach is not mathematically sound and therefore, as an absolute minimum, it must be checked that it has a negligible effect. "

is not correct –Âãonce expanded this is fine so long as only a single profile is used per

modelled grid cell. As such, I retract this comment, and please ignore it.

I hope this has not caused the authors or editor any undue problems or confusion.

---

## Referee Comment (RC3) · Anonymous Referee #2 · 18 Feb 2021

**General comments**

This manuscript uses satellite retrievals of CH4 from the TANSO instrument onboard GOSAT to optimize CH4 fluxes globally, but with a focus on South America and, specifically, the Amazon Basin and Brazil. The optimized CH4 flux for the Amazon Basin shows a positive trend, which is strongest in the wet season, which the authors suggest could mean that the trend is driven by wetland emissions. The authors find that grid-cells with a flux positive trend generally coincide with grid-cells with a positive temperature trend, but there is also some coincidence with grid-cells with decreasing wetland fraction. Overall, the methodology appears sound and the results are well presented. However, there are a few issues that should be addressed before publication. The main issues are outlined as follows:

The authors emphasize that the positive flux trend in the Amazon Basin is strongest in the wet season and suggest that this is likely a result of wetland emissions. However, the fraction of wetland area in the Amazon has also been decreasing over the same time period. Based on the data presented it is not possible to conclude anything about the cause of the trend, since it could be also due to an increase of biomass burning and/or agricultural emissions, which may be increasing more during the wet season than during the dry season. I think the authors should expand the discussion mentioning other possible causes of the trend and be clearer that with the data presented they cannot draw any conclusion about its cause.

Although the inversion was run globally, there is no mention of the global CH4 budget before and after the inversion. I think it is important to present the values of the global total source a priori and a posteriori, as well as the total calculated atmospheric sink. The global budget is especially relevant when discussing the Amazon emissions and their trend in the context of the global emissions.

**Specific comments**

L64: The authors could also cite Thompson et al. Geophys. Res. Lett. (2018) who like Worden et al. (2017) found an increase in both microbial and fossil fuel emissions. Thompson et al. (2018) also simultaneously optimized OH but found no significant OH trend.

L150-152: By "the GOSAT averaging kernels were averaged similarly to the XCH4" do the authors mean that the averaging kernels (AKs) of all retrievals falling within a single model time-step and grid-cell were averaged (as was XCH4)? I think this should be specified to avoid any confusion.

L155-166: The authors state that they compare CH4 mixing ratios, from a previous inversion using only the surface network of flask sampling sites, to GOSAT XCH4. They then fit a quadratic function to the observation-model differences as a function of latitude and add the calculated bias error to their prior modelled XCH4 in the inversion with GOSAT XCH4. I see one problem with this approach. That is, an offset between the optimized XCH4 and GOSAT XCH4 is expected since the information from the surface observations is limited, especially in the tropics (e.g. Fig. 1). Ignoring for the moment any atmospheric transport error, this means that the information from GOSAT (i.e. model-observation difference between the prior modelled XCH4 and GOSAT XCH4) is reduced.

L159: I think the authors should state here that they fitted a quadratic to the modelobservation error as a function of latitude.

L198-191: The authors should specify that INVICAT is an inversion framework which uses the forward and adjoint models of TOMCAT.

L233: For completeness, the authors should state what "scaled as in McNorton et al." means.

L235: The authors should state what "in a configuration described in McNorton et al." means.

L236: The authors should also state what scaling was applied to the rice emission estimates. Furthermore, rice emissions are already included in the anthropogenic emission estimate from EDGAR-v4.2FT2010. Was there a double counting of rice emissions in the prior estimate?

L236-237: The authors should specify what other natural sources were included in their prior emissions estimate.

L245-247: There are a number of recent studies addressing the possible trend in OH and OH variability related to ENSO (e.g. Zhao et al., Atmos. Chem. Phys. 2020 and Anderson et al., Atmos. Chem. Phys. 2020). The authors should mention the possible

СЗ

ENSO influence here.

L252: This sentence is a bit confusing, do the authors mean that the simple bottom-up model only provided a climatological (i.e. with no year to year variability)? Or do they mean that they used meteorological (and other) driving data? Later on (L278-279) it sounds as though it is the latter.

L286-287: What do the authors mean by "we consider only the wet season NAT + AGR + BB emissions ... which we assume to be almost entirely from wetlands"? The BU model described is only for wetland emissions, so I think AGR + BB are in fact ignored?

L378: The authors should specify what they mean by "performance", i.e. mean bias and correlation.

Table 2: In the caption, by "optimal" the authors mean the "better" statistic is in bold. I suggest changing "optimal" to "better" or similar, as "optimal" could be confused with being from the optimization.

L381-382: In fact the posterior correlation is better for observations <1.5 km at all sites except ALF. Only the bias increases (more positive) for all sites, except SAN.

L457-461: The INVICAT results used to optimize the BU model are the total of the sectors "NAT+AGR+BB", while the BU model only considers wetland emissions. Therefore, is it realistic to think that the BU model can reproduce the trend or variability seen the INVICAT results, even when only the wet season emissions are considered? In other words, is the assumption that the emissions during the wet season are dominated by wetlands reasonable? Could the positive trend seen in INVICAT "NAT+AGR+BB" which appears to be approximately spatially correlated with a positive temperature trend (and with a negative wetland fraction trend) be driven by biomass burning or agricultural emissions rather than wetland emissions? I think it is not possible to draw any conclusions about which sector is driving the positive trend based on the data presented here. Fig. 8: I suggest changing the title of (c) to "NAT+AGR+BB flux trend" as in the caption.

L458: By "curve fitting program" I think the authors mean a multiple linear regression was used to determine the values of q10, a1 and a2, I think this should be stated more clearly. Also, with only 3 variables, and given that the model is only for wetland emissions, whereas the observation, i.e. the INVICAT result, is for NAT+AGR+BB, it is somehow to be expected that the optimized BU model cannot reproduce the INVICAT result.

L544: Related to the above comments, I don't think the authors can conclude that wetland emissions are likely driving the positive trend based only on the fact that the trend is strongest during the wet season. I think there needs to be more analysis of the possibility of biomass burning emissions increasing during the wet season, and trends in agricultural emissions.

Technical comments

L125: should be "Data from these sites are assimilated..."

L341: I think this should be "...emissions in Brazil are nearly constant over time..." and not "consistent"

L588: should be "...a period during which there was widespread flooding."

**C5**

---

## Referee Comment (RC4) · Anonymous Referee #3 · 20 Feb 2021

The author uses the GOSAT satellite-based columnar and surface observation of CH4 to optimize CH4 emissions for 2010-2018, focusing on South America, especially the Amazon river basin and Brazil. The author reported an increasing emission trend for optimizing emissions over Brazil. The author further observed the strong trend during the wet season and attributed the wetland emissions as the main driver. The author used the bottom-up model to investigate the causes of variations in wetland CH4 emissions. However, the fluxes neither match with variation in annual fluxes nor the positive trend in the inversion emissions. The author reported no change in fossil fuel emissions for the same period. Overall, the results are impressive, with detailed and careful investigations. The paper would greatly benefit from a full revision to ensure the authors'

key messages are easier to understand. Below I outline my substantive and minor comments.

Substantive Comments:

1) The paper looks lengthy, which make it difficult to understand key messages clearly. I would suggest revising the structure and moving the proxy information (e.g., Detailed of Bottom-up simulations) in the supplement part, as indicated by Luke Western.

2) Estimated error covariance is as essential as estimated fluxes, yet these are never shown. I would like to know what the decrease in error covariance is between the prior and posterior.

3) The accuracy of model transport plays a crucial role in interhemispheric gradients in CH4 near the Earth's surface and vertical gradient in the troposphere and stratosphere since the total column consists of 40% of stratospheric air, depending on seasons and latitude. Thus, it is crucial to test the accuracy of the troposphere and stratosphere vertical transport before optimization. Too fast transport of too slow transport in the model will directly hinder the optimization results. The vertical gradient in the stratosphere becomes more critical for simulating XCH4 data. Along with transport, the validation of chemical loss is also vital for optimizations. It would be useful if the authors reported the methyl chloroform e-folding lifetime as a way of assessing the prior OH and another proxy (SF6) simulations for validating the inter-hemispheric and vertical transport. Even if this is addressed in an earlier paper, some Supplementary or acknowledgement plots would be useful.

4) It is difficult to follow how authors get the uncertainty ranges (e.g., Fig. 4a, b). It would be generous to the reader if you state clearly this information in the text.

5) It is currently difficult to understand the spatial distribution of sectoral emissions to understand their role in the study region's methane emission changes. The spatial distribution of prior emissions (probably Figure in the supplementary Section), mostly like

wetland emissions, Enteric fermentation and manure management emissions, biomass burning, and Fossil Fuel, will help the reader.

6) It would be not easy to pinpoint the role of wetland emission in increasing trend compared to the decreasing wetland areas for the contemporary period. In such a case, the role of bottom-up simulations is challenging to connect the dots. What about the role of Enteric fermentation and manure management, and agricultural emission, which are also increasing over Brazil as per the updated version of EDGAR inventory (EDGARv4.3 and latest version)? Investigating such dimension will make the study interesting. The spatial sectorial emission map of or trend in the prior sectorial emissions can help excavate such information.

7) Using the histogram for the validation does not give enough information about how well the optimized emissions improved the fitting of the observed trend? It would be useful for the reader to show the observed and fitted XCH4 time-series (say over the region shown in Fig. 5) and the time series over two different altitude range of aircraft as shown in the Figure. 6.

Specific comments. L32: "Cannot match..." –> "Neither match...". Since you are using "nor" conjunction L31: "Much of CH4...." Consider giving the numerical fraction. L43: "remove "to the atmosphere" and "in the atmosphere". Can state like "However, the magnitude of global sources and sinks are still not well quantified..."

L50: The period is not matching with the previous studies.

L64: The authors could also cite Chandra et al. JMSJ. (2021) (https://www.jstage.jst.go.jp/article/jmsj/advpub/0/advpub_2021-015/_article), who also found an increase in both biogenic and fossil fuel emissions. Naus et al (2020) (https://doi.org/10.5194/acp-2020-624) and Patra et al. (2021) (https://agupubs.onlinelibrary.wiley.com/doi/10.1029/2020JD033862) also studied the OH trend using CH3CCl3 observations and observed no significant trend.

L58: Point 1st and 3rd have more or less the same conclusion. Consider it to merge. So basically, you will have two points of source uncertainty and sink uncertainty. L64: Better to state more conclusively. Worden et al. (2017) suggest that the equal contribution of both FF and biogenic sources is also possible if pyrogenic emissions are decreased for the same period.

Section 2.1.3. It is not mentioned in this Section the use of aircraft measurements in this study. After reading this paragraph, I thought the aircraft measurements are also used in optimization—the information regarding the use of these observations for the validation purpose is coming later.

L215. Not clear 250% of what? Is uncertainty 250% of prior emissions?

Section2.2.2 Consider showing all the sectoral emissions time series for the study region (maybe in Supplementary?)

L304. "The posterior residuals show no significant trend or seasonality" -→ Over where? Over Amazon Basin or the whole of South America?

L305. Similar emission maps for each sector in Supplementary will help the reader to understand the dominant emission sources over different regions.

L335. Figure S1 is not shown. Maybe you are talking about Figure A1. Correct others also throughout the manuscript. L345. How did you calculate the uncertainty? L346. "This means….." How does the previous sentence follow this conclusion? at least gives the number…. Consider reformulating

L353. What is the shoulder season?

---

## Author Comment (AC1) · 1 Apr 2021

The manuscripts presents estimates of methane emissions, focused on Brazil and the Amazon Basin, using a top-down inversion approach, which is validated against independent measurements, and compared to a bottom up model. The manuscript is thorough and detailed, which will undoubtedly be informative for future studies. The conclusions are supported by the findings of the study, which are suitably appraised. As such, I hope to see its eventual publication in ACP. There is however, a major error in the methodology – as it is stated in the paper – which may or may not have a large impact on the results. This must be addressed before the manuscript can be considered acceptable for publication. In addition, below are a number of suggestions for revisions to improve the manuscript, followed by technical comments.

We thank the reviewer for his comments, which have significantly helped to improve the paper and clarify our results and message. We hope that we have addressed these concerns appropriately. We note this reviewer has withdrawn his comment regarding the major error in the methodology (see below and RC2). Our point-by-point response is given below, highlighted as blue text.

1) Line 150-151: "For accurate comparison between the retrieved XCH4 and those simulated by the model, the GOSAT averaging kernels were averaged similarly to the XCH4 and applied to the model vertical profiles." This approach is not mathematically sound and therefore, as an absolute minimum, it must be checked that it has a negligible effect. Otherwise it could lead, quite likely, to an underestimation of the modelled XCH4 and thus overestimate emissions. The reason for the error here is as follows. If we simplify the maths to just two variables, and let $X$ be the concentration at each of the $i$ the model levels, and $A$ be the diagonal of the averaging kernel, and $Y$ be the XCH4, then we can say that $Y = \sum_i a_i x_i$. If we take the mean of each observation $j$ of the total $n$ observations in a grid cell then we want $Y_{grid} = \frac{1}{n} \sum_j Y_j$, which is equal to $\frac{1}{n} \sum_j \sum_i a_{ij} x_{ij}$. As there is only one $X$ per grid cell in the model, we can simplify this to $\frac{1}{n} \sum_j \sum_i a_{ij} x_i^{grid}$. From the text, it states that instead the average of the averaging kernel was applied to the modelled profile, so $Y_{grid} = \sum_i \frac{1}{n} \left( \sum_j a_{ij} \right) x_i^{grid} = \frac{1}{n} \sum_i x_i^{grid} \sum_j a_{ij}$, which is not equivalent. Please let me know if this needs more clarification.

Although the reviewer has since retracted this comment (see below and RC2), we have clarified our reasoning for using this methodology. We have added text within the manuscript at line 247 as follows:

"Using a single model profile in each grid cell and model time step allows the use of averaging kernels that have been averaged in this way without introducing a bias, due to the distributive property of matrix multiplication."

We thank the reviewer for bringing up this comment and for being open and responsive in further discussions regarding this method, leading to improvements to the text.

2) Line 197: Although each has completed 40 iterations, how do you check that the optimisation routine has converged within 40 iterations?

We have clarified this in the text as follows:

"The inversions were carried out for each year separately and each completed 40 minimisation iterations. For each year's inversion, 40 iterations were enough for the cost function and its gradient norm to be judged to have converged (less than 1% variation through 5 consecutive iterations)."

3)  The paper is thorough and detailed, although – as a matter of opinion – it is in places quite arduous to read. I suggest making use of supplementary material and moving some of the analysis here. For example, at line 533: This section should be moved to the supplement, and referred to in line 531. The application of the B-U model as a whole and its discussion would be better placed in the supplement to improve the conciseness of the manuscript. This could also apply to the validation against the site and aircraft data – refer to the outcomes in the main text but the details can be moved to a supplement. This should get across the key points of the paper, i.e. the emissions and their sources, better to the reader.

We have generally attempted to shorten the main text wherever possible but have also moved some sections to the supplement as suggested here. The section on the bottom-up model has moved to Appendix B, whilst the details of the inversion at the higher resolution has moved to Appendix C. We have left the analysis of the independent validation in the main text, however.

4)  Code and data availability: It would be much more beneficial to the community if the results of this work (posterior emissions estimates in space and time) were publicly available. E.g. placing the spatial maps for the mean posterior emissions for Brazil and the posterior emissions estimates for Brazil each year in a public repository in e.g. netcdf format. I see that this has also been suggested by the topical editor, but I would like to reiterate its importance.

This is correct and we have since uploaded our prior and posterior monthly $CH_4$ fluxes for South America in netcdf format to the data archive of the Centre for Environmental Data Analysis (CEDA, https://www.ceda.ac.uk). The data can now be cited as follows:

Wilson, C.; Chipperfield, M.; Gloor, M.; Parker, R.; Boesch, H. (2021): Posterior South American monthly mean surface flux of methane (2010-2018) produced using the INVICAT 4D-Var inverse model. Centre for Environmental Data Analysis, *17 February 2021*. doi:10.5285/88224a922439441fa6644b4564dcd90c.

We have added this information to the manuscript.

**Technical comments:**

Throughout: Units for ACP should be expressed in exponential form, i.e. Tg yr−1 and not Tg/yr.

We have adapted our terminology as suggested.

Throughout: Be consistent with Tg(CH4)/yr and Tg/yr (e.g. line 81). I would recommend stating explicitly in its first occurrence that Tg/yr refers to Tg(CH4) and from thereon just writing Tg/yr.

We have adapted our terminology as suggested.

Throughout: There is an error in the nomenclature used throughout the manuscript. The manuscript often refers to the prior/posterior mole fraction when referring to a single parametric estimate; the prior/posterior are distributions of values. A better usage is a priori and a posteriori, or more explicit, the prior/posterior mean.

This is a good point; the former nomenclature is a little careless. Instead of correcting every instance of this through the text, which we believe reduces readability, we have inserted the following text in Section 2.2.1:

"The inversion input is in the form of an *a priori* mean flux value for each grid cell along with an error covariance matrix for these values, and the output is made up from an *a posteriori* mean grid cell flux value and error covariance matrix. Mean *a priori* and *a posteriori* atmospheric mole fractions of $CH_4$ are also produced. For ease, throughout the remainder of this text, we will refer to the mean values of the *a priori* and *a posteriori* fluxes as 'prior fluxes' and 'posterior fluxes', respectively. Similarly, the mean *a priori* and *a posteriori* mole fractions will be referred to as prior and posterior mole fractions."

Line 74: Basin is capitalised (not consistent with earlier use of basin).

We have changed all uses of 'basin' to lower-case for consistency.

Line 75: This sentence is confusing: it reads as though fires contribute to the number of wetland sources. Consider splitting this sentence as e.g. "as well as a number of other wetland sources in S America, emissions from. . ..also contribute to methane emissions."

We have changed this as suggested.

Line 77: Overlap in which sense? The emissions processes? In space? In stochastic error?

We have changed the text to clarify that flux contributions that we now know to be through trees is not extra contribution to the total but has likely already been counted as part of the wetland flux in some inventories.

Line 79: Consider "variability" rather than "variance", so as not to confuse with statistical variance.

Yes, this has been fixed.

Line 79: 'Earlier estimates' is inexact. Is this when the research was carried out, or the emissions from the year(s) in question? Specify the time period that you are discussing.

We have been more exact in our language here – "In studies published in the 2000s and early 2010s...".

Line 142: '. . .and found that the two agreed within their respective errors'. This sentence is meaningless without a description of the probability content in which these dataset agree (e.g. the 1 s.d. uncertain regions for both datasets overlap).

We have clarified this, explaining that 'biases between the satellite retrievals and the flask observations were not significantly different from zero'.

Eq(1): Why are the numbers in bold, as well as the brackets? Please remove.

Neither the numbers nor the brackets are in bold.

Line 170: Specify that it is the inset of Figure 1.

Done.

Line 171: 'until 2014' should not be in parenthesis.

Done.

Line 172: Until when? Present?

Yes, we have now clarified this.

Line 178: Space needed between 500 and m

Done.

Line 190: It is worth mentioning the species here, considering it is only 4 citations.

Done.

Line 194: Are these 5.6 degree square grids? Horizontal is vague.

Yes, we have clarified this.

Line 196: Use the latex command \citep[ERA-I][]{dee_reference} here.

We have altered this citation.

Line 215: They are given 250

We have corrected all instances of 'were given'.

Line 222: What was the functional family of the spatial correlation imposed? The most defensible choice here is a Matérn covariance structure (see Stein 2012, Interpolation of Spatial Data: Some Theory for Kriging), although it seems that this is not the case here.

We think that we previously used slightly unclear language in this section, and we have now clarified. Where we previously referred to the normal distribution of the uncertainty, which could refer to the underlying assumptions of the inversion methodology and the quadratic cost function, we now state:

"Both NAT + AGR + BB and FF sectors had spatial correlations imposed between grid cells, based on Gaussian covariance functions with correlation length scales of 500km."

We hope that it is now clear that we have not imposed a covariance structure such as that suggested by the reviewer. Whilst a Matérn structure can be favourable when representing some natural correlation structures, many previous studies, similar to ours, have used Gaussian spatial error functions. Due to the short correlation length used in our study, relative to the coarse nature of our model grid, model grid cells in South America are usually correlated strongly only with cells immediately next to them. In our case using a Matérn structure would therefore change the imposed covariances only in a minor way. In the future at higher resolutions, we would consider other covariance structures such as that suggested.

Line 223: Again, probability content of this uncertainty needs defining.

See comment above.

Line 226: L-BFGS is a general minimisation routine, not a method to explicitly derive a covariance matrix in the context of uncertainty. This needs rewording.

Yes, we have reworded this as follows:
'…using output from the L-BFGS method that we employ to minimise the cost function…'

Line 228: The 'cost function' is introduced for the first time here, and needs expanding upon for readers unfamiliar with the method.

Agreed, we have added a brief description of the cost function at the end of the first paragraph of this subsection.

Line 229: This should be the other way round – the lack of off-diagonals would give smaller emissions uncertainty than expected when including off-diagonals i.e. $var(\sum_i (x_i)) = \sum_i (var(x_i)) + \sum_{i \neq j} cov(x_i x_j)$.

We have changed this.

Line 237: Again, use \citep[MeMo][]{ref}.

We have altered this citation.

Eq 2.: Brackets are bold, when they shouldn't be.

The brackets are not bold.

(Paragraph starting 274: This paragraph reads well and is explicit. Ideally much more of the paper should read like this.)

Okay, thank you. We have attempted to reproduce this style elsewhere.

Line 301: 'The posterior error-weighted mean residual model-satellite mismatch' needs defining. It's unclear what this is.

We have clarified this: "The posterior (prior) mean model-satellite mismatch, weighted by the observation uncertainty,"

Figure 2: Consider earlier comment about the use of the terms prior/posterior. Note that the red-blue colour bar is not colour blind friendly.

See previous comment concerning the prior/posterior terminology. We understand that a red-blue diverging colour palette is, in fact, colour blind friendly and is one of those suggested by the ColorBrewer 2.0 website, which is recommended by the journal's submission criteria for colour blind people.

Line 310: Is this the mean of the prior/posterior means?

Yes, see previous comment concerning the prior/posterior terminology.

Line 318: This should say the spatial distribution of the posterior mean.

Yes, see previous comment concerning the prior/posterior terminology.

Line 345: What correlation do you assume when you make the assumption that they are highly correlated between years?

By reporting the mean uncertainty, we implicitly assume a correlation of 1 between the years' uncertainties. We do this rather than choose a relatively arbitrary correlation value. We have stated this in the text.

Figure 5: These colours are very difficult to differentiate. I suggest revising the shadings/colours in the figures.

We have changed the shading in this figure.

Line 458: The curve fitting programme (note 'programme' if using British English) needs elaborating on. What is the programme? What curve does it fit?

We have added extra text into the Appendix regarding this algorithm. British English uses the spelling 'program' when referring to computer code. However, we have adapted to use the word 'procedure' instead.

Line 462: If talking about 'no significant trend', the (statistical) significance of this trend must be given.

We have now stated that $p = 0.36$.

Line 474: Does 'here' refer to Section 4? If so, this should be stated as such: "Section 4 also shows..."

We have changed this to 'Figure 8e also shows…'.

Line 551: Use \citep[e.g.][]{ref}

We have altered this citation.

Line 585: An extra closing bracket is present.

This has been corrected.

Line 586: Consider 'substantial' instead of significant.

Done.

Line 598: Which other models besides Bloom et al.?

Changed to 'another model'.

Appendices Figures: Consider making these colour blind friendly.

We understand that a red-blue diverging colour palette is, in fact, colour blind friendly and is one of those suggested by the ColorBrewer 2.0 website, which is recommended by the journal's submission criteria for colour blind people.

---

## Author Comment (AC2) · 1 Apr 2021

Thank you for this additional response. Please see our response to the original comment in AC1.

---

## Author Comment (AC3) · 1 Apr 2021

**General comments**

This manuscript uses satellite retrievals of CH4 from the TANSO instrument onboard GOSAT to optimize CH4 fluxes globally, but with a focus on South America and, specifically, the Amazon Basin and Brazil. The optimized CH4 flux for the Amazon Basin shows a positive trend, which is strongest in the wet season, which the authors suggest could mean that the trend is driven by wetland emissions. The authors find that grid-cells with a flux positive trend generally coincide with grid-cells with a positive temperature trend, but there is also some coincidence with grid-cells with decreasing wetland fraction. Overall, the methodology appears sound and the results are well presented. However, there are a few issues that should be addressed before publication.

We thank the reviewer for their comments, which have significantly helped to improve the paper and clarify our results and message. We hope that we have addressed these concerns appropriately. Our point-by-point response is given below, highlighted as blue text.

The main issues are outlined as follows:

The authors emphasize that the positive flux trend in the Amazon Basin is strongest in the wet season and suggest that this is likely a result of wetland emissions. However, the fraction of wetland area in the Amazon has also been decreasing over the same time period. Based on the data presented it is not possible to conclude anything about the cause of the trend, since it could be also due to an increase of biomass burning and/or agricultural emissions, which may be increasing more during the wet season than during the dry season. I think the authors should expand the discussion mentioning other possible causes of the trend and be clearer that with the data presented they cannot draw any conclusion about its cause.

Yes, this is true, and we have rewritten much of our discussion and conclusions in order to make this more clear. We have now stated throughout that we are not able to draw definitive conclusions regarding the cause of the increase and expanded our explanation of the possible causes to include biomass burning and agricultural emissions. We continue to include our analysis using a bottom-up wetland model, but now clarify that this model's poor performance in matching the inversion results could be due to non-wetland sources being responsible for the variations.

Although the inversion was run globally, there is no mention of the global CH4 budget before and after the inversion. I think it is important to present the values of the global total source a priori and a posteriori, as well as the total calculated atmospheric sink. The global budget is especially relevant when discussing the Amazon emissions and their trend in the context of the global emissions.

We agree that this is important and have now included the *a priori* and *a posteriori* global source values at the start and end of the study period within the main text. The prior values are in Section 2.2.2, whilst the posterior values are in Section 3.2. We have also now included the values of the total atmospheric sinks due to OH (494.5 Tg in 2009) due to O1D and Cl (19.5 Tg in 2009) and due to the soil sink (33.9 Tg in 2009).

**Specific comments**

L64: The authors could also cite Thompson et al. Geophys. Res. Lett. (2018) who like Worden et al. (2017) found an increase in both microbial and fossil fuel emissions. Thompson et al. (2018) also simultaneously optimized OH but found no significant OH trend.

We now have included this reference.

L150-152: By "the GOSAT averaging kernels were averaged similarly to the XCH4" do the authors mean that the averaging kernels (AKs) of all retrievals falling within a single model time-step and grid-cell were averaged (as was XCH4)? I think this should be specified to avoid any confusion.

Yes, we have now clarified this.

L155-166: The authors state that they compare CH4 mixing ratios, from a previous inversion using only the surface network of flask sampling sites, to GOSAT XCH4. They then fit a quadratic function to the observation-model differences as a function of latitude and add the calculated bias error to their prior modelled XCH4 in the inversion with GOSAT XCH4. I see one problem with this approach. That is, an offset between the optimized XCH4 and GOSAT XCH4 is expected since the information from the surface observations is limited, especially in the tropics (e.g. Fig. 1). Ignoring for the moment any atmospheric transport error, this means that the information from GOSAT (i.e. model-observation difference between the prior modelled XCH4 and GOSAT XCH4) is reduced.

We agree that this method could potentially lead to a reduction in accurate information gained from the GOSAT data, but we believe that it is the best way to simultaneously assimilate the surface observations along with the satellite data. Due to a combination of biases in the model transport and chemistry and the satellite retrievals, there is a persistent offset between the model and satellite even when the surface is well-constrained by flask observations alone. The latitudinal function that we include is an attempt to account for these persistent biases and removes the conflict between the two sets of observations. This method has been previously employed in similar studies (e.g. Bergamaschi et al. (2009)) The fact that the bias function varies only in latitude, and is constant longitudinally and in time, means that information content along these two axes is preserved. We have added a short amount of text to reinforce this message.

Bergamaschi, P., et al. (2009), Inverse modeling of global and regional $CH_4$ emissions using SCIAMACHY satellite retrievals J. Geophys. Res., 114, D22301, doi:10.1029/2009JD012287

L159: I think the authors should state here that they fitted a quadratic to the model-observation error as a function of latitude.

This is stated a little further down in this paragraph.

L198-191: The authors should specify that INVICAT is an inversion framework which uses the forward and adjoint models of TOMCAT.

Yes, we have now included this information.

L233: For completeness, the authors should state what "scaled as in McNorton et al." means.

We have added 'to apply an increasing global linear trend for the period after 2012'.

L235: The authors should state what "in a configuration described in McNorton et al." means.

We have added 'using four separate carbon pools to drive methanogenesis'.

L236: The authors should also state what scaling was applied to the rice emission estimates. Furthermore, rice emissions are already included in the anthropogenic emission estimate from EDGAR-v4.2FT2010. Was there a double counting of rice emissions in the prior estimate?

No, we did not include the EDGAR estimate of flux from rice agriculture, so there is no double-counting. We added the information that the scaling applied to the rice emissions was 0.75.

L236-237: The authors should specify what other natural sources were included in their prior emissions estimate.

Done.

L245-247: There are a number of recent studies addressing the possible trend in OH and OH variability related to ENSO (e.g. Zhao et al., Atmos. Chem. Phys. 2020 and Anderson et al., Atmos. Chem. Phys. 2020). The authors should mention the possible ENSO influence here.

We had missed these references and others, along with some discussion of ENSO-driven OH variability, and appreciate that they are important. We have now included this discussion here as suggested.

L252: This sentence is a bit confusing, do the authors mean that the simple bottom-up model only provided a climatological (i.e. with no year to year variability)? Or do they mean that they used meteorological (and other) driving data? Later on (L278-279) it sounds as though it is the latter.

Yes, we have changed this to 'meteorological and ecological input data'.

L286-287: What do the authors mean by "we consider only the wet season NAT + AGR + BB emissions . . . which we assume to be almost entirely from wetlands"? The BU model described is only for wetland emissions, so I think AGR + BB are in fact ignored?

This was poorly phrased, as we intended to say that we considered the NAT + AGR +BB posterior inversion flux, which we assume to be mainly from wetlands, for comparison with the B-U model output. We have clarified this in the text.

L378: The authors should specify what they mean by "performance", i.e. mean bias and correlation.

Yes, we have clarified this.

Table 2: In the caption, by "optimal" the authors mean the "better" statistic is in bold. I suggest changing "optimal" to "better" or similar, as "optimal" could be confused with being from the optimization.

Good point, we have changed this as suggested.

L381-382: In fact the posterior correlation is better for observations <1.5 km at all sites except ALF. Only the bias increases (more positive) for all sites, except SAN.

Yes, we have clarified this point.

L457-461: The INVICAT results used to optimize the BU model are the total of the sectors "NAT+AGR+BB", while the BU model only considers wetland emissions. Therefore, is it realistic to think that the BU model can reproduce the trend or variability seen the INVICAT results, even when only the wet season emissions are considered? In other words, is the assumption that the emissions during the wet season are dominated by wetlands reasonable? Could the positive trend seen in INVICAT "NAT+AGR+BB" which appears to be approximately spatially correlated with a positive temperature trend (and with a negative wetland fraction trend) be driven by biomass burning or agricultural emissions rather than wetland emissions? I think it is not possible to draw any conclusions about which sector is driving the positive trend based on the data presented here.

Please see our earlier response regarding this comment, which was generally applied to the whole discussion.

Fig. 8: I suggest changing the title of (c) to "NAT+AGR+BB flux trend" as in the caption.

Yes, we have changed this panel title.

L458: By "curve fitting program" I think the authors mean a multiple linear regression was used to determine the values of q10, a1 and a2, I think this should be stated more clearly. Also, with only 3 variables, and given that the model is only for wetland emissions, whereas the observation, i.e. the INVICAT result, is for NAT+AGR+BB, it is somehow to be expected that the optimized BU model cannot reproduce the INVICAT result.

We have now clarified the details of the curve-fitting program, and made it clear that if wetlands were not able to reproduce the INVICAT result, then it suggests that non-wetland sources could have been responsible. We have generally changed the previous tone of the text which suggested that wetlands were responsible for observed changes.

L544: Related to the above comments, I don't think the authors can conclude that wetland emissions are likely driving the positive trend based only on the fact that the trend is strongest during the wet

season. I think there needs to be more analysis of the possibility of biomass burning emissions increasing during the wet season, and trends in agricultural emissions.

Yes, see previous comments.

**Technical comments**

L125: should be "Data from these sites are assimilated. . ."

Done.

L341: I think this should be ". . .emissions in Brazil are nearly constant over time. . ." and not "consistent"

Done.

L588: should be ". . .a period during which there was widespread flooding."

Done.

---

## Author Comment (AC4) · 1 Apr 2021

The author uses the GOSAT satellite-based columnar and surface observation of $CH_4$ to optimize CH4 emissions for 2010-2018, focusing on South America, especially the Amazon river basin and Brazil. The author reported an increasing emission trend for optimizing emissions over Brazil. The author further observed the strong trend during the wet season and attributed the wetland emissions as the main driver. The author used the bottom-up model to investigate the causes of variations in wetland CH4 emissions. However, the fluxes neither match with variation in annual fluxes nor the positive trend in the inversion emissions. The author reported no change in fossil fuel emissions for the same period. Overall, the results are impressive, with detailed and careful investigations. The paper would greatly benefit from a full revision to ensure the authors' key messages are easier to understand. Below I outline my substantive and minor comments.

We thank the reviewer for his/her comments, which have significantly helped to improve the paper and clarify our results and message. We hope that we have addressed these concerns appropriately. Our point-by-point response is given below, highlighted as blue text.

**Substantive Comments:**

1) The paper looks lengthy, which make it difficult to understand key messages clearly. I would suggest revising the structure and moving the proxy information (e.g., Detailed of Bottom-up simulations) in the supplement part, as indicated by Luke Western.

We have generally attempted to shorten the main text where possible and have also moved some sections to the supplement as suggested here and by other reviewers.

2) Estimated error covariance is as essential as estimated fluxes, yet these are never shown. I would like to know what the decrease in error covariance is between the prior and posterior.

OK, we have included a plot of this within the Supplementary Material.

3) The accuracy of model transport plays a crucial role in interhemispheric gradients in CH4 near the Earth's surface and vertical gradient in the troposphere and stratosphere since the total column consists of 40% of stratospheric air, depending on seasons and latitude. Thus, it is crucial to test the accuracy of the troposphere and stratosphere vertical transport before optimization. Too fast transport of too slow transport in the model will directly hinder the optimization results. The vertical gradient in the stratosphere becomes more critical for simulating XCH4 data. Along with transport, the validation of chemical loss is also vital for optimizations. It would be useful if the authors reported the methyl chloroform e-folding lifetime as a way of assessing the prior OH and another proxy (SF6) simulations for validating the inter-hemispheric and vertical transport. Even if this is addressed in an earlier paper, some Supplementary or acknowledgement plots would be useful.

You are correct that model transport errors will likely play an important role in the results obtained in this work. We attempt to account for a significant proportion of model transport error, particularly in the vertical distribution, through addition of a model – satellite bias term derived from a previous inversion based on surface flask observations of $CH_4$, whilst acknowledging that there will very likely remain some model error that is unaccounted for. We have previously investigated the

simulated interhemispheric gradient (IHG) in TOMCAT ourselves through $SF_6$ simulations (e.g. Wilson et al. (2014)), and have also taken part in many multi-model intercomparisons, including for $CH_4$ (e.g. Patra et al. (2011), Thompson et al. (2014)). In all cases the model performs well, particularly in comparison with other similar models. In Patra et al. (2011), the TOMCAT model used the OH fields that we use in this study (TOMCAT $CH_4$ lifetime: 9.98 years), and also applied them to simulations of MCF, matching the observed MCF lifetime well (TOMCAT: 4.71 years compared to observed value of 4.9± 0.3 yr (Prinn et al., 2005)). However, it is true that the IHG in TOMCAT tends to be slightly overestimated compared to observations, and this will likely affect our results. We have added this caveat into the main text in the model description section and referred the reader to the references discussed here. We feel, however, that replications of figures from previous studies within this manuscript would only act to unnecessarily increase the length and complexity of this paper and hope that referring the reader to previous work is a satisfactory response.

4) It is difficult to follow how authors get the uncertainty ranges (e.g., Fig. 4a, b). It would be generous to the reader if you state clearly this information in the text.

This information is included in Section 2.2.1 and has been clarified.

5) It is currently difficult to understand the spatial distribution of sectoral emissions to understand their role in the study region's methane emission changes. The spatial distribution of prior emissions (probably Figure in the Supplementary Section), mostly like wetland emissions, Enteric fermentation and manure management emissions, biomass burning, and Fossil Fuel, will help the reader.

OK, we have included a plot of this within the Supplementary Material.

6) It would be not easy to pinpoint the role of wetland emission in increasing trend compared to the decreasing wetland areas for the contemporary period. In such a case, the role of bottom-up simulations is challenging to connect the dots. What about the role of Enteric fermentation and manure management, and agricultural emission, which are also increasing over Brazil as per the updated version of EDGAR inventory (EDGARv4.3 and latest version)? Investigating such dimension will make the study interesting. The spatial sectorial emission map of or trend in the prior sectorial emissions can help excavate such information.

Yes, as suggested by this reviewer and others, we have changed the tone of the main text to make it clear that wetlands are not the only potential culprit for changes in the total flux and that using our methodology does not allow us to further speculate on the source of the variations. We have included the reference to the latest EDGAR dataset, and have also included the possibility that biomass burning sources could be changing outside of the dry season. The trend in the prior emissions used in our study is negligible, so we decided not to include it as a figure.

7) Using the histogram for the validation does not give enough information about how well the optimized emissions improved the fitting of the observed trend? It would be useful for the reader to show the observed and fitted XCH4 time-series (say over the region shown in Fig. 5) and the time series over two different altitude range of aircraft as shown in the Figure. 6.

We have included a plot of observed (GOSAT) and simulated XCH4 over the Amazon basin and Brazil in the Supplementary material (Figure A4). The aircraft observations will be published in a separate upcoming manuscript.

**Specific comments.**

L32: "Cannot match. . ." –> "Neither match. . .". Since you are using "nor" conjunction

Done.

L31: "Much of CH4. . .." Consider giving the numerical fraction.

If this refers to line 39, we have adapted the text as follows:
"Approximately 90% of the $CH_4$ that is emitted into the atmosphere is eventually destroyed through reaction with the hydroxyl (OH) radical and the remainder is lost to other smaller sinks,…"

L43: "remove "to the atmosphere" and "in the atmosphere". Can state like "However, the magnitude of global sources and sinks are still not well quantified..."

Done.

L50: The period is not matching with the previous studies.

Yes, we have changed '2007' to '2006'.

L64: The authors could also cite Chandra et al. JMSJ. (2021) (https://www.jstage.jst.go.jp/article/jmsj/advpub/0/advpub_2021-015/_article), who also found an increase in both biogenic and fossil fuel emissions. Naus et al (2020) (https://doi.org/10.5194/acp-2020-624) and Patra et al. (2021) (https://agupubs.onlinelibrary.wiley.com/doi/10.1029/2020JD033862) also studied the OH trend using CH3CCl3 observations and observed no significant trend.

Thank you for highlighting these important recent references, we have included them in the text.

L58: Point 1st and 3rd have more or less the same conclusion. Consider it to merge. So basically, you will have two points of source uncertainty and sink uncertainty.

We have merged and slightly rewritten the two points.

L64: Better to state more conclusively. Worden et al. (2017) suggest that the equal contribution of both FF and biogenic sources is also possible if pyrogenic emissions are decreased for the same period.
We have added this information.

Section 2.1.3. It is not mentioned in this Section the use of aircraft measurements in this study. After reading this paragraph, I thought the aircraft measurements are also used in optimization. The information regarding the use of these observations for the validation purpose is coming later.

Yes, we have added text at the start of this section to clarify this.

L215. Not clear 250% of what? Is uncertainty 250% of prior emissions?

Yes, we have now included this information.

Section2.2.2 Consider showing all the sectoral emissions time series for the study region (maybe in Supplementary?)

We have included maps of the sectorial emissions. There is actually very little variation in time in the prior emissions, so we have decided not to include a figure showing the sectorial timeseries.

L304. "The posterior residuals show no significant trend or seasonality" -! Over where? Over Amazon Basin or the whole of South America?

We have clarified that residuals show no trend within either region.

L305. Similar emission maps for each sector in Supplementary will help the reader to understand the dominant emission sources over different regions.

We have included maps of the sectorial emissions in the Supplementary section.

L335. Figure S1 is not shown. Maybe you are talking about Figure A1. Correct others also throughout the manuscript.

Done.

L345. How did you calculate the uncertainty?

This is described in Section 2.2.1

L346. "This means. . ..." How does the previous sentence follow this conclusion? at least gives the number. . .. Consider reformulating

The sentence actually says 'this *mean* flux' but we accept that this could be unclear. We have changed it to 'Our mean flux value'.

L353. What is the shoulder season?

This ambiguous phrase has been removed.